# Intracellular calcium leak lowers glucose storage in human muscle, promoting hyperglycemia and diabetes

Eshwar R Tammineni[1], Natalia Kraeva[2,3], Lourdes Figueroa[1], Carlo Manno[1], Carlos A Ibarra[2,3], Amira Klip[4], Sheila Riazi[2,3], Eduardo Rios[1]*

[1]Department of Physiology and Biophysics, Rush University Medical Center, Chicago, United States; [2]Malignant Hyperthermia Investigation Unit (MHIU) of the University Health Network (Canada), Toronto, Canada; [3]Department of Anaesthesia & Pain Management, Toronto General Hospital, UHN, University of Toronto, Toronto, Canada; [4]Cell Biology Program, The Hospital for Sick Children, Toronto, Canada

**Abstract** Most glucose is processed in muscle, for energy or glycogen stores. Malignant Hyperthermia Susceptibility (MHS) exemplifies muscle conditions that increase $[Ca^{2+}]_{cytosol}$. 42% of MHS patients have hyperglycemia. We show that phosphorylated glycogen phosphorylase (GP*a*), glycogen synthase (GS*a*) – respectively activated and inactivated by phosphorylation – and their $Ca^{2+}$-dependent kinase (PhK), are elevated in microsomal extracts from MHS patients' muscle. Glycogen and glucose transporter GLUT4 are decreased. $[Ca^{2+}]_{cytosol}$, increased to MHS levels, promoted GP phosphorylation. Imaging at ~100 nm resolution located GP*a* at sarcoplasmic reticulum (SR) junctional cisternae, and *apo*-GP at Z disk. MHS muscle therefore has a wide-ranging alteration in glucose metabolism: high $[Ca^{2+}]_{cytosol}$ activates PhK, which inhibits GS, activates GP and moves it toward the SR, favoring glycogenolysis. The alterations probably cause these patients' hyperglycemia. For basic studies, MHS emerges as a variable stressor, which forces glucose pathways from the normal to the diseased range, thereby exposing novel metabolic links.

*For correspondence:
erios@rush.edu

Competing interests: The authors declare that no competing interests exist.

## Introduction

Skeletal muscle is the major processing site for dietary glucose, consuming it at a high rate during exercise and storing it as glycogen more than any other tissue (*Baron et al., 1988*; *Mizgier et al., 2014*). Muscle is critical to glucose homeostasis, to the point that its failure to take up glucose heralds the eventual onset of type 2 diabetes (*DeFronzo and Tripathy, 2009*). Reciprocally, glucose is required to provide fuel necessary for muscle contraction and thermogenesis. This relationship consists of a two-way mutual control process: postprandial secretion of insulin increases muscle uptake of glucose, and multiple muscle-generated signals increase glucose uptake during exercise or thermogenesis.

Changes in cytosolic $Ca^{2+}$ concentration ($[Ca^{2+}]_{cyto}$) play a central —albeit not fully understood— role in these interactions. Insulin induces intracellular $Ca^{2+}$ release sufficient for a modest, transient increase in $[Ca^{2+}]_{cyto}$ (e.g. *Contreras-Ferrat et al., 2014*; *Lanner et al., 2008*; *Park et al., 2015*); reciprocally, signaling from contractile activity mediated by increased $[Ca^{2+}]_{cyto}$ results in increased glucose uptake by insulin-dependent and independent mechanisms (*Jessen and Goodyear, 2005*; *Lanner et al., 2009*; *Holloszy and Narahara, 1967*; *Holloszy et al., 1986*). Once inside the myofiber, whether glucose is processed for direct energy production or stored as glycogen is tightly dictated by contractile activity and energy demand. Because proper function of muscle requires maintenance of ATP concentrations within a narrow range, the myofiber maintains spatially

**eLife digest** Animals and humans move by contracting the skeletal muscles attached to their bones. These muscles take up a type of sugar called glucose from food and use it to fuel contractions or store it for later in the form of glycogen. If muscles fail to use glucose it can lead to excessive sugar levels in the blood and a condition called diabetes. Within muscle cells are stores of calcium that signal the muscle to contract. Changes in calcium levels enhance the uptake of glucose that fuel these contractions. However, variations in calcium have also been linked to diabetes, and it remained unclear when and how these 'signals' become harmful.

People with a condition called malignant hyperthermia susceptibility (MHS for short) have genetic mutations that allow calcium to leak out from these stores. This condition may result in excessive contractions causing the muscle to over-heat, become rigid and break down, which can lead to death if left untreated. A clinical study in 2019 found that out of hundreds of patients who had MHS, nearly half had high blood sugar and were likely to develop diabetes. Now, Tammineni et al. – including some of the researchers involved in the 2019 study – have set out to find why calcium leaks lead to elevated blood sugar levels.

The experiments showed that enzymes that help convert glycogen to glucose are more active in patients with MHS, and found in different locations inside muscle cells. Whereas the enzymes that change glucose into glycogen are less active. This slows down the conversion of glucose into glycogen for storage and speeds up the breakdown of glycogen into glucose. Patients with MHS also had fewer molecules that transport glucose into muscle cells and stored less glycogen. These changes imply that less glucose is being removed from the blood.

Next, Tammineni et al. used a microscopy technique that is able to distinguish finely separated objects with a precision not reached before in living muscle. This revealed that when the activity of the enzyme that breaks down glycogen increased, it moved next to the calcium store. This effect was also observed in the muscle cells of MHS patients that leaked calcium from their stores. Taken together, these observations may explain why patients with MHS have high levels of sugar in their blood.

These findings suggest that MHS may start decades before developing diabetes and blood sugar levels in these patients should be regularly monitored. Future studies should investigate whether drugs that block calcium from leaking may help prevent high blood sugar in patients with MHS or other conditions that cause a similar calcium leak.

distributed glycogen deposits, together with the enzymatic machinery needed to rapidly process glycogen towards glycolysis (*Adeva-Andany et al., 2016*; *Ørtenblad and Nielsen, 2015*). The mechanisms that integrate muscle activity and glucose processing rely on changes in $[Ca^{2+}]_{cyto}$ during excitation-contraction coupling. The present study addresses the $Ca^{2+}$-dependent control of the balance between glycogen synthesis and glycogenolysis (*Ozawa, 2011*).

The close interactions between glucose metabolism and calcium signaling become relevant in type 2 diabetes (*Contreras-Ferrat et al., 2014*; *Park et al., 2015*; *Guerrero-Hernandez and Verkhratsky, 2014*). The diabetes-induced failure in mechanisms that regulate cell $Ca^{2+}$ (*Levy, 1999*) usually results in an increase in $[Ca^{2+}]_{cyto}$. Because of many possible mutual interactions between $Ca^{2+}$ and glucose transport, it is not clear whether the increase in $[Ca^{2+}]_{cyto}$ is a contributor to the origin of the disease (*Park et al., 2015*; *Zarain-Herzberg et al., 2014*), nor whether it is a cause or consequence of hyperglycemia (*Lanner et al., 2008*). In any case, the alteration in $[Ca^{2+}]_{cyto}$ will in turn modify the pathogenic processes of diabetes (*Contreras-Ferrat et al., 2014*).

Malignant Hyperthermia Susceptibility (MHS) (*Litman et al., 2018*) is a condition usually linked to mutations in *RYR1*, which encodes the $Ca^{2+}$ release channel of the sarcoplasmic reticulum (SR), or in other genes that encode proteins of the skeletal muscle couplon (*Stern et al., 1997*; *Franzini-Armstrong et al., 1999*). The primary defect underlying this condition is believed to be a '$Ca^{2+}$ leak' from the SR, due to an excessive propensity of the RyR channel to open, which in turn leads to increased cytosolic $[Ca^{2+}]$. Large increases in $[Ca^{2+}]_{cyto}$ have been found in muscle from MHS individuals (*López Padrino, 1994*; *Lopez et al., 1992*), MH-susceptible pigs (*Lopez et al., 1986*) and in myotubes derived from MHS patients (*Figueroa et al., 2019*).

An MHS-like phenotype occurs in multiple muscle diseases; the estimates of its prevalence vary widely depending on methods and assumptions (with results as high as 1/200 [*Bachand et al., 1997*] and as low as 1/50000 [*Ording, 1985*]). MHS is clinically diagnosed through measurements of the magnitude of contractile forces ($F_H$ and $F_C$) developed by freshly biopsied fiber bundles upon exposure to stimulants halothane and caffeine, standardized as the Caffeine Halothane Contracture Test (CHCT *Larach, 1989*) and the In Vitro Contracture Test (*Hopkins et al., 2015*). Recently Altamirano et al. observed a high prevalence of hyperglycemia in a large group with positive CHCT in our clinic (*Altamirano et al., 2019*), attributing this association to the increase in $[Ca^{2+}]_{cyto}$ previously found in MHS patients (*López Padrino, 1994*).

These observations are indicators of strong underlying connections between excitation-contraction signaling and glucose metabolism. This study was undertaken to understand these connections and derive consequences for patient management, through the investigation of the quantity, phosphorylation and cellular location of the enzymes that control glycogen storage and utilization, in muscle cells from MHS patients. The findings establish Malignant Hyperthermia Susceptibility and the diseases presenting this condition as prodromes and contributors to the development of hyperglycemia and eventually type 2 diabetes (e.g. *Diabetes Canada Clinical Practice Guidelines Expert Committee, 2020*). Additionally, they provide new insights into the normal glucose processing pathways of skeletal muscle.

## Results

### MHS patients have elevated glucose

560 patients (named the 'legacy' cohort) were subjected to the CHCT and studied clinically in our Malignant Hyperthermia Investigation Unit (MHIU) from 1994 to 2013; 329 of them were diagnosed as MHS ($F_{H} >0.6$ g and/or $F_{C} >0.2$ g) and 231 as MHN. Their levels of fasting glucose (measured within the last 3 years) are plotted in *Figure 1* vs. the increase in force of their stretched muscle bundles under exposure to 3% halothane. As reported (*Altamirano et al., 2019*), and shown in the box plot (1B), 40.0% (132/329) had elevated fasting blood sugar (FBS, here used as synonymous with 'glycemia'), which is more than double the expected prevalence in the age-matched general population. Additionally, plot 1A shows a positive and highly significant correlation between FBS and $F_H$ ($r = 0.21$, $p$ of no correlation <0.001). Consistent with the correlation, the median FBS (illustrated in 1B) was significantly higher in MHS (5.4 mM, Inter Quartile Range 4.9–6.9) than in MHN (5.1 mM, IQR 4.8–5.7). FBS was not significantly different between women and men but it was significantly correlated with age (Panel 1C) and Body Mass Index (BMI, 1D). The 1st order regression lines of $F_H$ and age are nearly parallel for MHN and MHS, separated by ~0.5 mM, which indicates that the MHS condition associates with elevated FBS in an age-independent manner. Panel E graphs the numbers of patients in the four age bins delimited by the green lines. Black segments represent numbers of MHN subjects with normal glycemia; white segments those with hyperglycemia. The red-tone bars correspondingly represent the MHS numbers. In 1F the numbers are represented as fractions of the total in each bin. The graphs show that the prevalence of hyperglycemia increases with age and is higher in MHS than in MHN at any age.

Moreover, the Diabetes Canada Clinical Practice Guidelines Expert Committee stipulates four criteria, any one of which is sufficient for the diagnosis of diabetes (*Diabetes Canada Clinical Practice Guidelines Expert Committee, 2020*). According to criterion #1 (FBS $\geq 7.0$ mmol/L), 22% of the patients in the 'legacy' cohort are diabetic, which more than doubles the prevalence in the general age-matched Canadian population. The balance of comparisons answers in the affirmative the question posed by *Altamirano et al., 2019*, establishing MHS as a prodrome of hyperglycemia and a path to diabetes.

Our study seeks cell-level mechanisms that may explain the high prevalence of hyperglycemia in the 'legacy' cohort. It focuses on a 'recent' group of subjects diagnosed since 2014 to date, whose biopsies are systematically subjected to studies in the lab at Rush University. By contrast with the 'legacy' group, in the 'recent' cohort the incidence of hyperglycemia and the correlation between FBS and $F_H$ are lower (3.5%, $r = 0.04$, *Figure 1—figure supplement 1*); the differences could be ascribed to an average age 24 years less than that of the 'legacy' cohort. (Within the 'recent' cohort, the MHN group is considered a suitable substitute for a sample of healthy individuals. These are

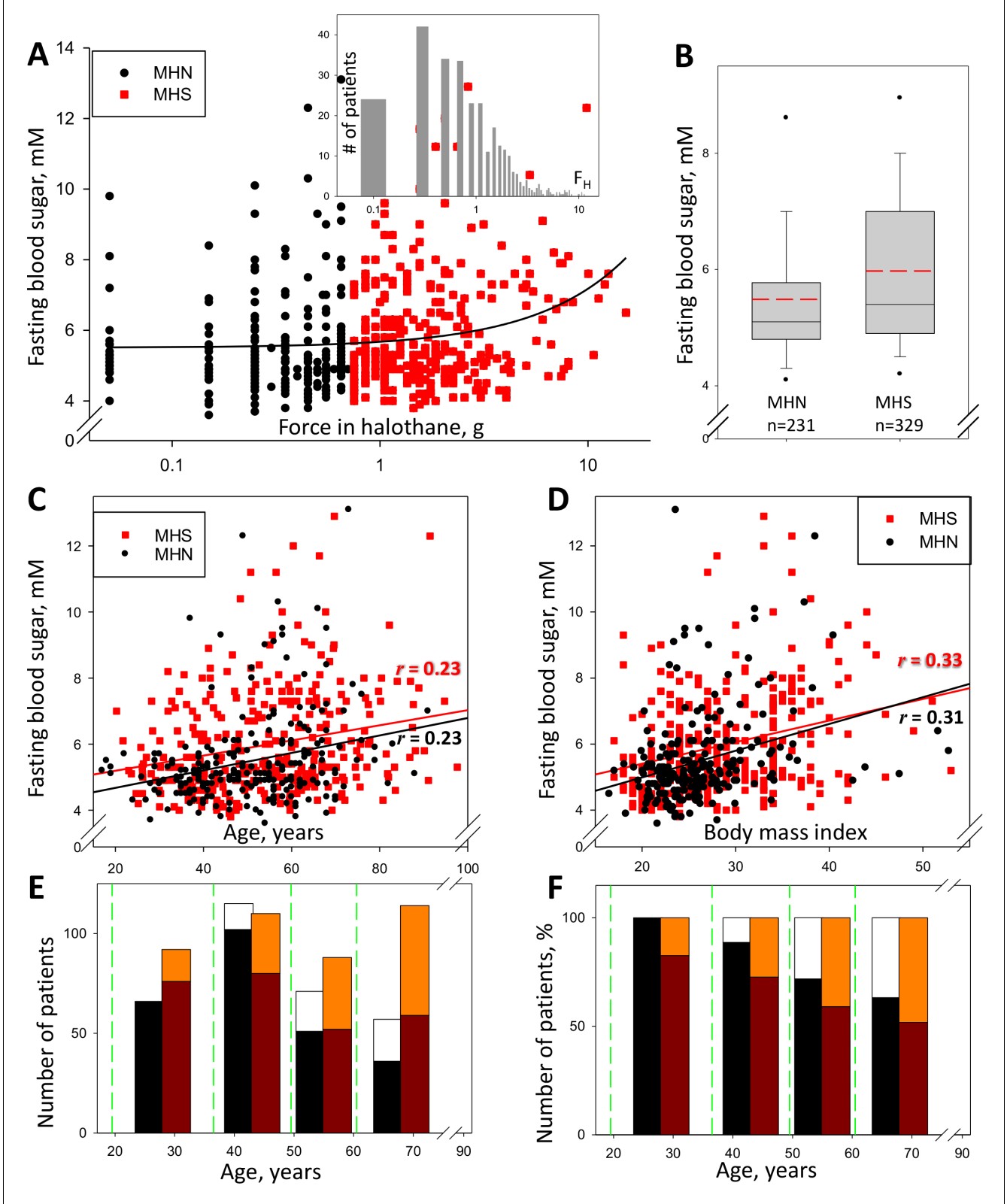

**Figure 1.** Glycemia and susceptibility to Malignant Hyperthermia. (A) $F_H$ (muscle force in response to 3% halothane) vs. FBS in 560 subjects studied between the years 2009 and 2013 (the 'legacy cohort'). Data for those with $F_{Hx00A0} > 0.6$ g, deemed susceptible to MH ('MHS'), are represented in red. The linear best fit (solid line) yields a regression coefficient (slope) of 0.17 mM/g; the correlation coefficient $r = 0.21$, $p$ of no correlation is < 0.001. Inset, distribution of $F_H$ in all patients. (B) Box plot of data in A. The means (S.E.M.) in mM, are 5.49 (0.090) for MH-negatives ('MHN') and 5.95 (0.088) for
*Figure 1 continued on next page*

*Figure 1 continued*

MHS; in a rank-sum (u) test, *p* of no difference is < 0.001. (C) FBS correlation with age. *r* = 0.23 and *p* is < 0.001 for both MHN and MHS; slopes are similar, 0.026 and 0.023 mM/yr respectively, which reflects an age-independent effect of the MHS condition. (D) FBS correlation with BMI. For MHN, *r* = 0.31; for MHS, *r* = 0.33; p < 0.001 for both. The regression coefficient is higher for MHN than MHS (0.081 and 0.066 mM/unit respectively). (E) Number of patients in four age bins delimited by the green lines. Black/white bars represent MHN and red-tone bars MHS individuals. Number of subjects with FBS $\geq$ 6 mM are represented by the upper segment of the bars. (F) Percentage of the total number in each bin and CHCT class. The fraction of patients with high FBS increases with age but the excess in MHS persists.

The online version of this article includes the following figure supplement(s) for figure 1:

**Figure supplement 1.** Response to halothane vs. FBS in members of the 'recent' cohort.

subjects tested only because of family history, who additionally have proven negative to relevant gene mutations. None of them has signs of muscle disease.)

## The content of GP and GDE is greater in MHS microsomal fractions

The muscles of MHS patients had major changes in protein endowment. This is illustrated in *Figure 2B*, in a Ponceau-stained gel of the microsomal fraction of biopsies of 12 MHS and 13 MHN patients selected as described in Materials and methods. Greater density of two bands near 100 and 180 kDa was visible in the microsomal fraction (arrows), but not in whole muscle lysates from the same patients (*Figure 2J*, see also *Figure 2—figure supplement 1*). Studied by mass spectrometry (Appendix 1), the 100 kDa and 180 kDa bands contain respectively and in high abundance glycogen phosphorylase (GP) and glycogen debranching enzyme (GDE).

The quantity of GP and other proteins was determined by Western blotting as illustrated in *Figure 2*. All blots were quantified by a custom application, described in Materials and methods and demonstrated with a recorded analysis session presented as *Video 1* (see also Materials and methods for a formal expression of the linear relationship between band signal and protein content and *Figure 2—figure supplement 2* for its validation). The antibody used in *Figure 2A* reacts with both the phosphorylated GP (GP*a*) and its *apo* form (GP*b*). The protein detected is referred to as *all-forms* GP, or just GP. The content of GP derived from this blot, equal to the sum of GP*a* and GP*b* contents, was greater by 119% in the MHS group (p < 0.001; panel 2F and *Table 1*).

Glycogenolysis requires GP operating together with glycogen debranching enzyme (GDE). Both proteins are found associated with glycogen granules, which share the intermyofibrillar space with the SR and are extracted with the microsomal fraction (*Ørtenblad and Nielsen, 2015*; *Fridén et al., 1989*). GDE, the main component of the band at 180 kDa, is visibly increased in the gels of MHS samples. The GDE blot (2C) derived from the gel in 2B confirms the impression, showing a mean content increased by 94% (p = 0.002). Patients that had higher GP had a commensurate elevation in GDE (*r* = 0.94, panel 2H). The procedure to derive Western blots for multiple proteins from the same electrophoresis gel is illustrated with *Figure 2—figure supplement 3*.

The differences in GP and GDE can also be quantified directly on the gels stained nonspecifically, without resorting to Western blots. A comparison of the GP band for 13 MHN and 12 MHS muscles in one gel yields an excess of 73% in MHS (p = 0.005) and a similar difference between the GDE bands (p = 0.002, *Table 1*). In the gel of *Figure 2B* the band at 100 kDa has the greatest signal mass in the whole gel; the excess protein in this band of the MHS (73%) is comparable with that of GP measured with protein-specific antibodies. The excellent correlation (*r* = 0.92) between the band signal in Western blots and in the stained gel is demonstrated in *Figure 2—figure supplement 4*. Taken together, the observations identify GP as the main constituent of the band and the most abundant protein in the microsomal fraction.

The differences in these and other glycogen processing enzymes described later are in contrast with the invariance of two couplon proteins tested (calsequestrin one or Casq1, and FKBP12, lower panels in *Figure 2*) and a third calcium handling SR protein, SERCA1 (p=0.65; *Table 1* and *Figure 2—figure supplement 5*). Remarkably therefore, in these subjects bearing a primary calcium handling defect, the expression levels are altered more for two glycogen breakdown enzymes than any of the major calcium management proteins studied here.

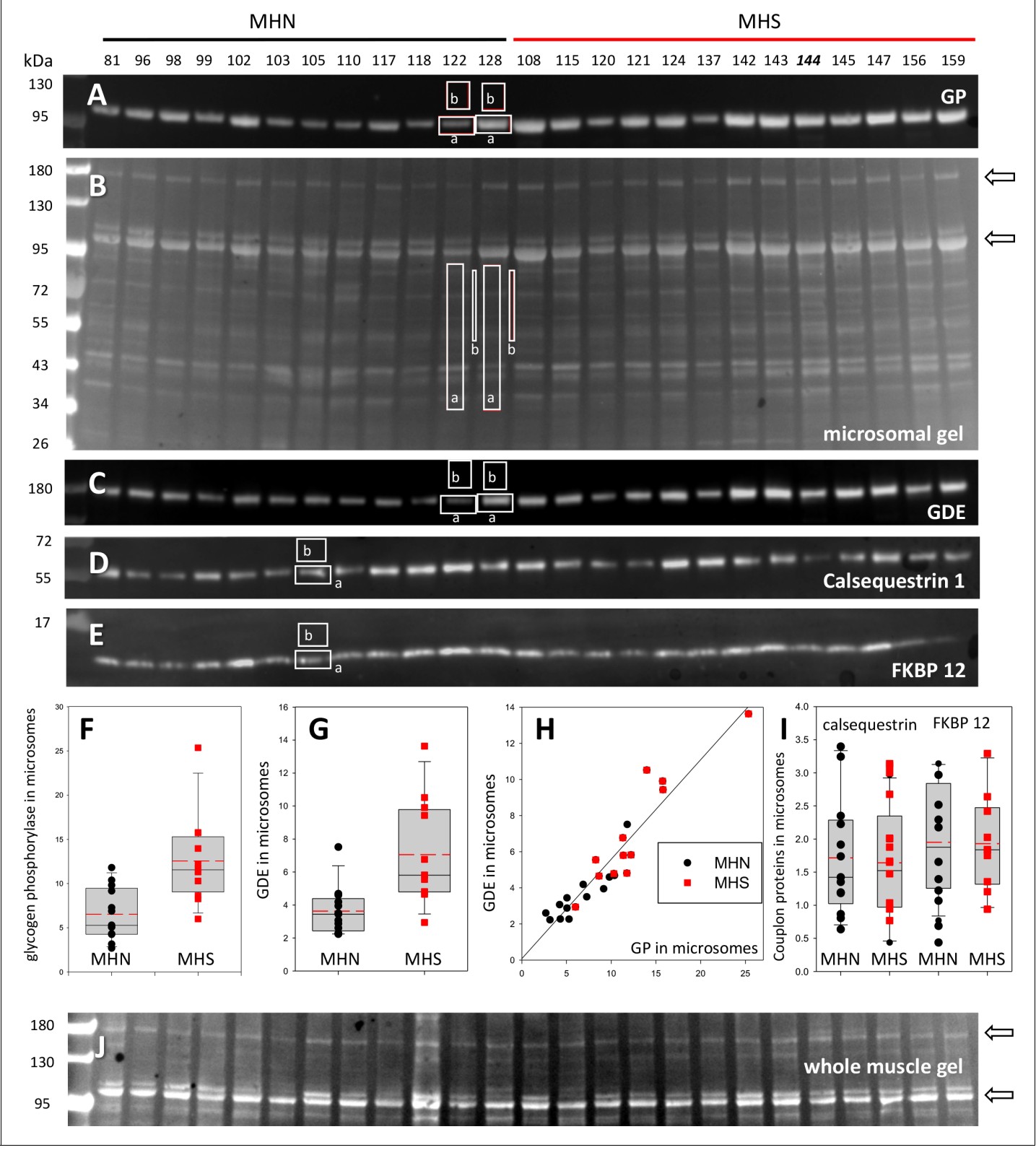

**Figure 2.** Protein content in patients' microsomes. (**A**) Western blot analysis of glycogen phosphorylase (GP) for 25 subjects, in a luminescence scale with black as 0 and white as saturating value. The numbers above each lane are patient identifiers that apply to all 25-lane gels shown in the article. The twelve lanes under the black bar have protein from MHN patients; those under the red bar are from MHS patients, except #144, which was reclassified as MHN. (**B**) Ponceau-stained gel that originated all blots in the figure. A and B illustrate our custom quantitative analysis. The content of every protein

*Figure 2 continued on next page*

*Figure 2 continued*

quantified in blots was calculated as the signal mass within region a above background (average level in b). Content was normalized for quantity of preparation in the lane dividing by the signal similarly calculated in the gel in B. The signal in B is computed in a large area of the lane to average multiple proteins in the fraction. The method is fully demonstrated in **Video 1**. Arrows in B mark two bands, near 100 and 180 kDa, with a visibly greater signal in the MHS group. Their main components were GP and glycogen debranching enzyme (GDE) respectively. (**C–E**) Western blots of GDE, calsequestrin 1 and FKBP12 in different sections of gel B (identified by molecular weight markers at left). (**F and G**) Box plots of GP and GDE content. In both cases the content was greater on average for MHS and the differences significant (data in **Table 1**). (**H**) GP vs. GDE in blots A and C; $r = 0.94$; $p < 10^{-4}$. (**I**) Comparing content of calsequestrin 1 and FKBP12. The differences are not significant (**Table 1**). (**J**) Detail of a whole tissue gel of the same patients, to show lack of visible differences between MHS and MHN in region of GP and GDE (arrows). The contrast is highly increased to show bands. Full gel in whole tissue is in **Figure 2—figure supplement 1**; proportionality of signal and protein quantity in **Figure 2—figure supplement 2**, the technique to derive blots for multiple proteins from the same gel in **Figure 2—figure supplement 3**. Direct quantification of proteins in the gel in **Figure 2—figure supplement 4** and the distribution of SERCA1 in microsomes in **Figure 2—figure supplement 5**.

The online version of this article includes the following figure supplement(s) for figure 2:

**Figure supplement 1.** Glycogen Phosphorylase in whole tissue extract.
**Figure supplement 2.** Relationship between blot signal and quantity of protein.
**Figure supplement 3.** Procedure to derive blots for multiple proteins from the same gel.
**Figure supplement 4.** Direct quantification of GP in Ponceau-stained gel.
**Figure supplement 5.** The distribution of SERCA 1.

## The phosphorylation of GP increases in the whole muscle extract

GP is phosphorylated by phosphorylase kinase (PhK) (**Newgard et al., 2000**). PhK, GP, GDE and glycogen synthase (GS), together with GP phosphatase PP1, and initiator and scaffolding proteins (**Crosson et al., 2003**), associate with glycogen into glycogen granules (**Prats et al., 2018**). Phosphorylation at Ser14 converts GP from the *apo* or *b* form into GP*a*, which promotes the active conformation of the enzyme (**Nagy, 2017**). In human and murine muscle, glycogen granules are largely inter-myofibrillar (**Ørtenblad and Nielsen, 2015**), close to the SR.

Given that PhK is activated by $Ca^{2+}$ (**Brushia and Walsh, 1999**) and assuming that MHS patients' $[Ca^{2+}]_{cyto}$ is elevated (see Introduction), we hypothesize that the excess $[Ca^{2+}]_{cyto}$, via promotion of PhK, leads to an increase in GP*a*. To justify the observation of excess GP in microsomes, we also hypothesize that GP*a* (rather than GP*b*) migrates with microsomes, presumably incorporating into glycogen granules. The quantities of *all-forms* GP (i.e. GP*a* and GP*b*) and GP*a* were measured in immunoblots from different gels loaded with aliquots of the same extracts (**Figure 3**). In the whole tissue, GP was greater in the MHS by 29% (p=0.07), GP*a* by 68% (p = 4 10⁻⁴, **Figure 3A,B**) and the ratio GP*a*/GP by 42% (p = 0.005, **Figure 3D**). In the microsomal fraction GP*a* content was 70% higher in the MHS (p = 0.05); because *all-forms* GP also increased (**Figure 2**), the ratio of contents GP*a*/GP did not change significantly (p = 0.28, **Figure 3H**). If all the GP in microsomes were phosphorylated, an increase in GP*a* would not change the ratio GP*a*/GP. This possibility is supported by results that follow.

To recapitulate, the MHS have more GP*a* in their muscle and elevated GP in microsomes. As shown in **Figure 3E**, the two variables are positively correlated regardless of MH status ($r = 0.72$, p = 0.01, **Table 2**) — patients who have high phosphorylation of GP in the whole muscle generally have high GP content in microsomes. These observations suggest a simple explanation: GP*a* is largely in the intermyofibrillar space, associated with the SR, hence in biochemical analyses it will appear largely with the microsomal fraction. Adding to this assumption the hypothesis raised previously, that all GP in microsomes is phosphorylated, it follows that the greater content of GP in the microsomal fraction of MHS muscle is due to the increase in GP*a*. The combined hypotheses (GP*a* is extracted together with the microsomal fraction

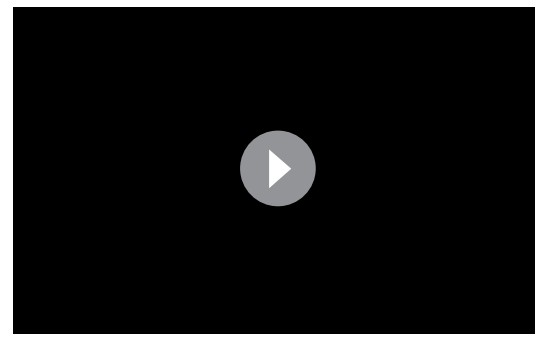

**Video 1.** Demonstration of a custom method to quantify Western blots.
https://elifesciences.org/articles/53999#video1

**Table 1.** Muscle contents of proteins and glycogen, and glucose in blood.

mc – microsomal fraction; wm – whole muscle lysate. Rows 1–17 list statistical parameters for 13 MHN and 12 MHS subjects. Units for glycogen are g/g of protein in extract, mM for FBS and arbitrary for all other variables (Materials and methods). Row 18 lists parameters of fasting blood sugar in 78 MHN and 80 MHS patients of the recent cohort. In row 19 FBS parameters of 6 'metabolically normal' and 6 'metabolically challenged' patients (*MN* and *MC*, defined in Discussion) are listed in columns 3–5 and 6–8, respectively. § Significant at the 0.05 level by 2-tailed *t* test or *Mann-Whitney *u* test. Rows listing significant increases are colored red, reductions are in blue. The inclusion of two variables with p>0.05 is justified based on their correlations listed in *Table 2*.

| | 1 | 2 | 3 | 4 | 5 | 6 | 7 | 8 | 9 |
|---|---|---|---|---|---|---|---|---|---|
| | | | MHN | | | MHS | | | |
| | Species | Fraction | Mean | Median | S.E.M. | Mean | Median | S.E.M. | *P* |
| 1 | GP | mc | 6.53 | 5.28 | 0.813 | 12.6 | 11.6 | 1.440 | 0.001* |
| 2 | | wm | 3.55 | 3.33 | 0.198 | 4.10 | 4.28 | 0.208 | 0.07 |
| 3 | GP*a* | mc | 3.81 | 3.09 | 0.522 | 6.50 | 4.76 | 1.22 | 0.05§ |
| 4 | | wm | 1.32 | 1.26 | 0.090 | 2.22 | 2.08 | 0.198 | <0.001* |
| 5 | GP*a*/GP | mc | 0.63 | 0.55 | 0.082 | 0.51 | 0.44 | 0.074 | 0.28 |
| 6 | | wm | 0.39 | 0.36 | 0.061 | 0.54 | 0.51 | 0.065 | 0.005* |
| 7 | PhK | mc | 0.85 | 0.82 | 0.070 | 1.57 | 1.41 | 0.180 | <0.001§* |
| 8 | | wm | 0.81 | 0.82 | 0.101 | 1.06 | 0.81 | 0.169 | 0.57* |
| 9 | GDE | mc | 3.63 | 3.44 | 0.402 | 7.05 | 5.81 | 0.902 | 0.002* |
| 10 | GS | mc | 6.95 | 6.10 | 1.29 | 5.27 | 5.12 | 0.666 | 0.26 |
| 11 | GS*a*/GS | mc | 1.03 | 0.76 | 0.157 | 1.39 | 1.37 | 0.150 | 0.11 |
| 12 | GLUT4 | mc | 1.18 | 1.23 | 0.189 | 0.62 | 0.54 | 0.105 | 0.01§ |
| 13 | Glycogen | mc | 1.17 | 1.15 | 0.116 | 0.75 | 0.75 | 0.066 | 0.01* |
| 14 | Casq1 | mc | 1.71 | 1.42 | 0.249 | 1.64 | 1.52 | 0.261 | 0.84 |
| 15 | FKBP12 | mc | 1.95 | 1.88 | 0.227 | 1.93 | 1.84 | 0.223 | 0.95 |
| 16 | SERCA1a | mc | 18.5 | 18.9 | 1.365 | 17.4 | 19.2 | 1.728 | 0.61 |
| 17 | SERCA1b | mc | 7.71 | 8.34 | 0.839 | 7.18 | 7.23 | 1.019 | 0.69 |
| 18 | FBS, recent cohort | | 5.10 | 5.10 | 0.150 | 5.44 | 5.35 | 0.207 | 0.19 |
| 19 | FBS, *MN* or *MC* | | 5.05 | 5.05 | 0.256 | 5.90 | 5.85 | 0.241 | 0.03§* |

and all GP in microsomes is GP*a*) predict that GP*a* will be located at or near the SR. The prediction — which does not distinguish between locations of GP*a* in intermyofibrillar glycogen granules or bound to the SR — was tested in the patients' tissue by immunohistochemistry.

### The sarcomeric location of GP in human muscle

GP in its two forms was imaged in muscles of known content of *all-forms* GP and GP*a*. Thin, slightly stretched bundles were fixed in paraformaldehyde and differentially stained with two antibodies, one specific for GP*a* and one reactive to both forms. The immunofluorescence was acquired as *z*-stacks and visualized after deblurring (Materials and methods) either as individual images from the stacks (a.k.a. sections $F(x, y)$) or as 3-D reconstructions of the corrected stack. The image $F_{GP}(x, y)$ of the *all-forms* fluorescence was variable among different individuals and different myofibers from the same subject. As shown in *Figure 4*, GP (panel Aa) was sometimes present at highest density near the Z disk, where it colocalized with actinin-1 (Ab), but in most cases (panels 4 B) it occupied a slightly wider region, always in the interior of the sarcomeric I band, where most mitochondria are located (Bb, see inset for structural guidance). As is visually apparent and was confirmed by quantifying colocalization, GP and the mitochondrial marker formed I-band patches that remained largely separate. An illustrative study on living cells is in Panels C: images of an intact muscle fiber of an adult mouse expressing GFP-tagged GP, co-stained with TMRE, a marker of polarized mitochondria. *All-forms* GP (panel Ca) is distributed similarly as in the fixed human cells, largely sharing the I band with mitochondria (Cb) but without actual overlap (Cc). Confirming the separation, Western blots

**Table 2.** Correlations between contents (proteins and glycogen) in all patients.

mc – microsomal fraction; wm – whole muscle lysate. Numbers above diagonal are Pearson correlation coefficients $r$; below diagonal is $p$ of no correlation. Significant positive correlations highlighted in red, significant negative correlations in blue. Some of the correlations of GLUT4 and glycogen with the enzyme contents are not significant in binary tests, but acquire significance in multivariate tests (Appendix 3). Contents of SERCA 1 (and Casq1 and FKBP12, not shown) had no significant correlation with the enzymes studied.

| | | GP | | GPa | | PhK | | GSa/GS | GDE | GLUT4 | Glycogen | Serca 1 |
|---|---|---|---|---|---|---|---|---|---|---|---|---|
| | | Mc | Wm | Mc | Wm | Mc | Wm | Mc | | | | |
| GP | mc | | 0.41 | 0.72 | 0.64 | 0.83 | 0.52 | 0.53 | 0.94 | −0.46 | −0.47 | −0.24 |
| | wm | 0.02 | | 0.38 | 0.52 | 0.49 | 0.69 | 0.36 | 0.34 | −0.25 | −0.66 | −0.30 |
| GPa | mc | $<10^{-4}$ | 0.06 | | 0.64 | 0.75 | 0.55 | 0.52 | 0.48 | −0.51 | −0.36 | 0.10 |
| | wm | $<10^{-3}$ | <0.01 | $<10^{-4}$ | | 0.66 | 0.58 | 0.38 | 0.43 | −0.46 | −0.48 | −0.09 |
| PhK | mc | $<10^{-4}$ | 0.01 | $<10^{-4}$ | <0.01 | | 0.57 | 0.48 | 0.42 | −0.23 | −0.47 | −0.14 |
| | wm | <0.01 | $<10^{-3}$ | $<10^{-3}$ | <0.01 | <0.01 | | 0.37 | 0.49 | −0.06 | −0.21 | −0.27 |
| GSa/GS | mc | <0.01 | 0.08 | <0.01 | 0.06 | <0.01 | 0.07 | | 0.37 | −0.41 | −0.14 | −0.19 |
| GDE | | $<10^{-4}$ | 0.09 | 0.02 | 0.03 | 0.04 | 0.01 | 0.07 | | −0.09 | −0.09 | −0.26 |
| GLUT4 | | 0.02 | 0.21 | 0.01 | 0.02 | 0.25 | 0.77 | 0.04 | 0.66 | | 0.39 | 0.23 |
| Glycogen | | 0.02 | $<10^{-3}$ | 0.07 | 0.02 | 0.02 | 0.31 | 0.48 | 0.67 | 0.05 | | 0.15 |
| SERCA1 | | 0.24 | 0.11 | 0.63 | 0.67 | 0.48 | 0.19 | 0.34 | 0.20 | 0.26 | 0.75 | |

failed to find GP in mitochondrial fractions derived from human biopsies (*Figure 4—figure supplement 1*). Also against association with mitochondria are the variable distribution of GP (all-forms), which seems inconsistent with any systematic association, and the tentative placement of GP*b*, described below.

## Locating the apo form of GP

The panels in *Figure 4D* compare the all-forms fluorescence (4 Da) with that of the phosphorylated form (4 Db) in muscle of a patient with a high GP*a*/GP ratio. In muscles with high GP*a* content, the phosphorylated component was visible in the all-forms GP image as a pale 'shoulder', outside a more intense, single or dual central band. In these cases it was possible to derive a putative image of GP*b*, by subtracting from the *all-forms* image that of GP*a* multiplied by an adjustable factor $B/C$ that eliminated the GP*a* shoulder (Materials and methods *Equation 5*). Panel 4Dc is an overlay of the image of GP*a* and the image of GP*b* calculated thus (green); GP*b* appears as a dual (or sometimes single) filament, near the center of the I band. This analysis was completed in muscles of 10 patients, 5 MHN and 5 MHS, with similar results. $B/C$ varied between 0.2 and 0.8. The distribution in the longitudinal ($x$) direction was wider for GP*a*, as seen best in the $y$-averaged profiles $F(x)$ of panel 4E. The full width at half maximum of profiles $F_{GPa}(x)$ was on average 1.26 µm (S.E.M. 0.04), and that of $F_{GPb}(x)$, 0.85 (0.04) µm (p < $10^{-4}$). There were no significant differences between MHN and MHS. The putative images of GP*b* indicate that the GP detected in abundance in the microsomal fraction is mostly, if not entirely, the active (GP*a*) form.

The imaging study is consistent with the biochemical analyses; indeed, the strong correlation found between the content of *all-forms* GP in microsomes and its degree of phosphorylation in whole muscle is expected if GP in microsomes is phosphorylated. Likewise, the invariance of GP*a*/GP content in microsomes of MHN and MHS is consistent with the calculated distribution of GP*b* if all GP in microsomes is phosphorylated. The evidences from biochemical analyses and imaging jointly indicate that all or nearly all of the GP associated with the SR is phosphorylated. This inference was confirmed and the insight refined with high resolution imaging of GP*a*.

## GPa locates at or near SR terminal cisternae

To define precisely the location of GP*a*, myofibers of patients were co-stained with markers of SR sub-compartments and imaged in conditions designed to attain the highest spatial resolution achievable with conventional confocal fluorescence microscopy (Materials and methods). Panels A and B in

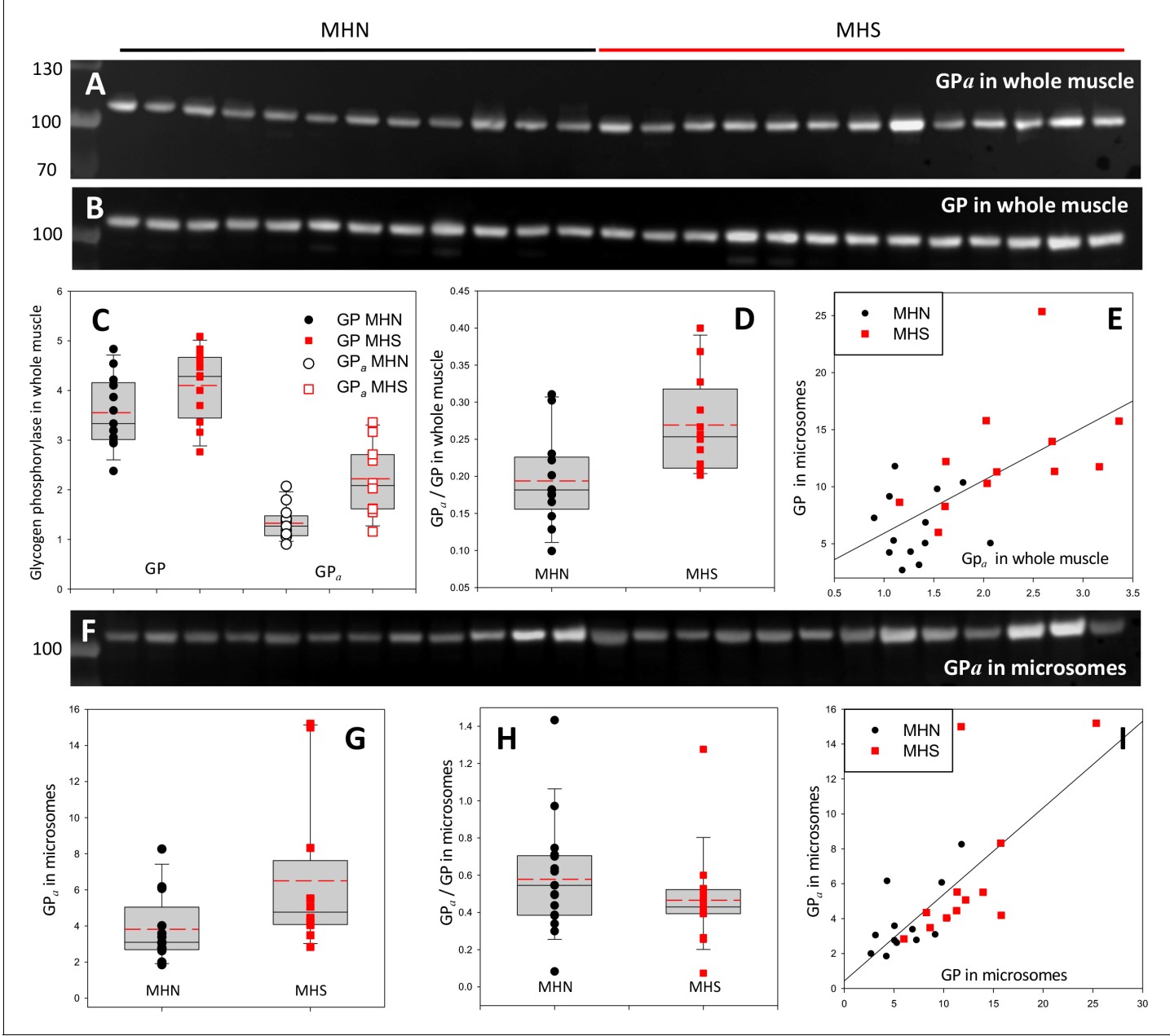

**Figure 3.** Phosphorylation of glycogen phosphorylase. (**A, B**) Western blots of 13 MHN and 12 MHS whole tissue lysates, identified in **Figure 2**. Blots were derived from different gels loaded with aliquots of the same preparations. Originating gels for this blot and others are in **Figure 3—figure supplement 1**. (**C**) The distribution of band signals in blots A and B. Statistical measures are listed in **Table 1**. GP (*all-forms*) and GP*a* were higher in the MHS (p=0.07 for GP and 3 10$^{-4}$ for GP*a*). (**D**) The median content ratio GP*a*/GP was 42% higher in the MHS (p = 0.005). (**E**) Cross-plot demonstrating correlation between GP*a* content in whole muscle and GP in microsomes (*r* = 0.64, p = 5 10$^{-4}$). (**F**) Western blot of GP*a* in the microsomal fractions studied in **Figure 2**. (**G, H**) Distributions of GP*a* in blot F and of the content ratio GP*a*/GP (the *all-forms* GP content was determined with blot A of **Figure 2**). GP*a* was higher by 71% (p = 0.05) but the phosphorylation ratio did not change significantly in the microsomal fraction (p = 0.28). (**I**) GP*a* vs. GP in microsomes. The correlation is high and highly significant (*r* = 0.72, p < 10$^{-4}$.).

The online version of this article includes the following figure supplement(s) for figure 3:

**Figure supplement 1.** Protein gels for blots.

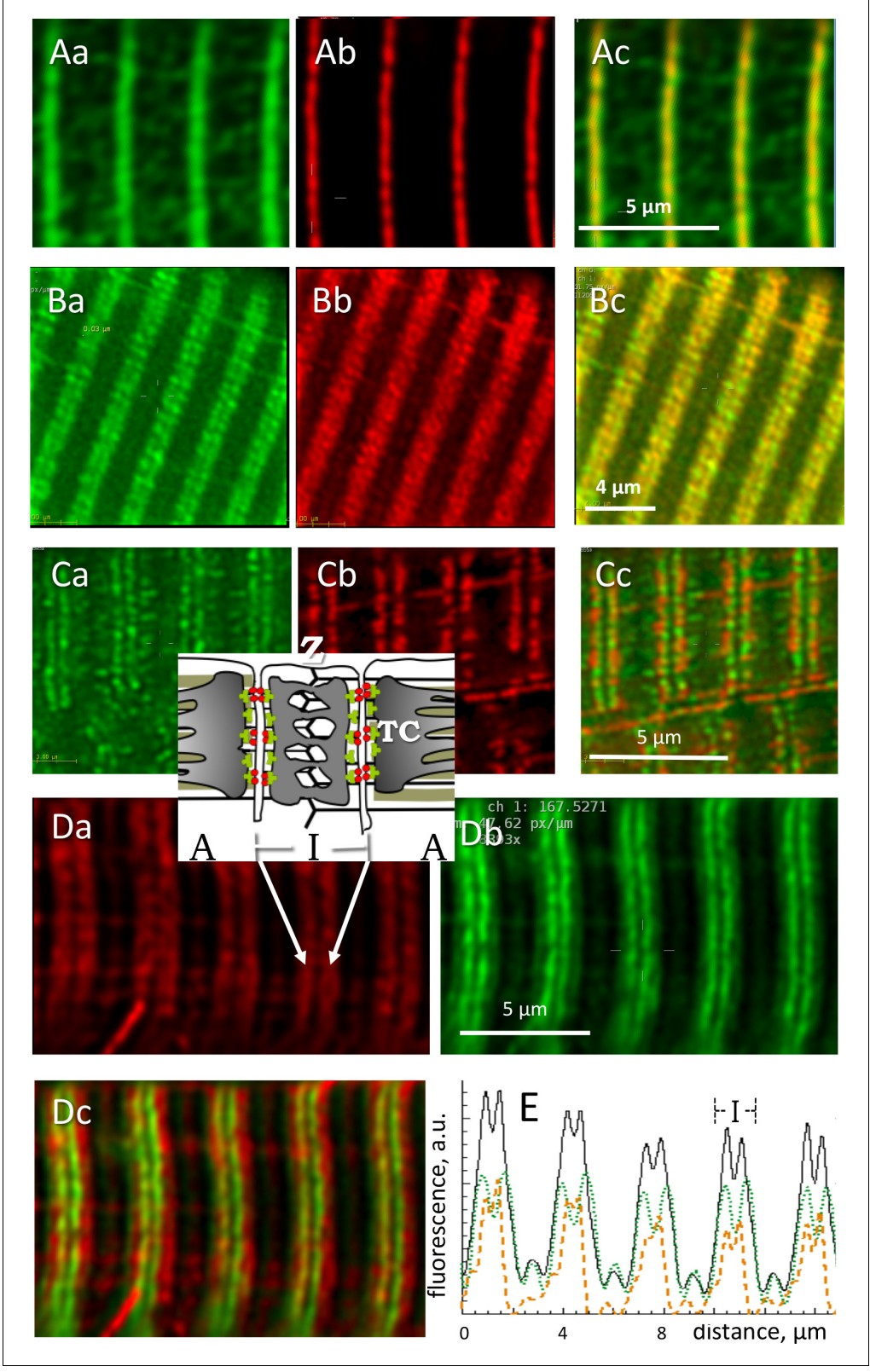

**Figure 4.** Locating phosphorylated and *apo* forms of glycogen phosphorylase. Structure elements are diagrammed in inset. (**A**) Images of all-forms GP (Aa), Z disk component α-actinin (Ab) and their overlay (Ac) in a biopsy from an MHN subject (# 139). Shown are individual images ('sections') from separate emission ranges ('channels') in a 2-channel *z*-stack, after correction for blurring (Materials and methods). GP largely overlaps with

*Figure 4 continued on next page*

*Figure 4 continued*

actinin and is seen as low intensity patches at other locations. (B) All-forms GP (Ba), cytochrome c oxidase subunit 4 (Cox4), located in mitochondria (Bb) and overlay (Bc). Both proteins are located largely in the sarcomeric I band, in patches with limited overlap. (C) Myocyte from adult mouse *FDB* expressing GP-GFP (Ca), co-stained with mitochondrial marker TMRE (Cb). Most GP is in the I band, without actual overlap with TMRE (Cc). No GP is detected in the mitochondrial fraction extracted from these muscles (*Figure 4—figure supplement 1*). (D) Human muscle cell co-stained for GP*a* and GP. Inset: location of Z disc and sarcomere bands I and A; TC, terminal cisternae of SR. GP*a* (Da) is distributed more broadly. Panel Dc shows in green a putative image of GP*b* (the *apo* form) obtained by subtracting from the all-forms GP fluorescence that of GP*a*, scaled by a factor that removes the 'shoulder' visible in the all-forms GP (*Equation 5*, Materials and methods). (E) The calculation is illustrated with image profiles F(x); black: $F_{GP/italic}$, green: $F_{GPa/italic}$, orange: $F_{GPb/italic}$. Panel Dc demonstrate the large difference in locations of GP*a* and GP*b*.

The online version of this article includes the following figure supplement(s) for figure 4:

**Figure supplement 1.** Absence of GP in mitochondrial fractions.

---

*Figure 5* show images of GP*a* (Aa, Ba) and co-stained Casq1 (Ab, Bb), a protein largely restricted to terminal cisternae (TC). In A are individual sections from *z* stacks, derived after correction for optical blurring, and in B, 3-D renderings of the entire stacks. The images resolve the two TC in a triad as separate bands of fluorescence. The separation, reported here for the first time using conventional fluorescence microscopy, is made possible by the improved spatial resolution (evaluated in figure supplements 1 and 2 of *Figure 5*). The separation of two bands of Casq1 is seen best in the 3-D rendering (Bb). The overlays Ac and Bc and the inset in Ac show that GP*a* is located close to Casq1, but without actual colocalization. (In these and other images, the two TC in a triad junction may be asymmetric, with more stain — be it Casq1 or GP*a* — in either the A-side or I-side TC. EM images of fixed muscle from our biopsies, recently acquired by Dr. Montserrat Samsó (Virginia Commonwealth University) frequently show asymmetric dilation of TC, which could explain the unequal fluorescence in the present images).

STIM1, an integral protein of the SR membrane, appears largely in the sarcomeric I band of mouse muscle (*Boncompagni et al., 2017*). In panels C and D, GP*a* delineates clearly the two TC in every triad and STIM1 forms small clusters. GP*a* and STIM1 can also be found outside triads, at less density, in locations consistent with the longitudinal SR (e.g., 5Dc). Again, there is close proximity but no actual colocalization between these proteins. A quantitative analysis of colocalization (illustrated with *Figure 5—figure supplement 1*) yields the average distance between nearest centers of signal mass in objectively located discrete GP*a* and STIM1 areas (*van Steensel et al., 1996*). The distance, 57 nm, is consistent with close juxtaposition of molecules within the diameter of a TC in the axial direction of the myofiber (*Franzini-Armstrong, 2018* and *Figure 5—figure supplement 1*).

The images also show STIM1 fluorescence inside a nucleus, a location first reported in cardiomyocytes (*Lee et al., 2018*) in putative association with nuclear invaginations of the endoplasmic reticulum (ER). This is the sole location where STIM1 and GP*a* were clearly apart, which suggests that the association of GP*a* and SR does not extend to the undifferentiated ER.

## The content of phosphorylase kinase (PhK) increases in MHS patients

To test whether the $Ca^{2+}$ activated kinase PhK provides the mechanistic link between the primary defect in MHS and the observed increased phosphorylation of GP, the PhK content of microsomes and whole cell extracts was compared between MHN and MHS subjects. Blots are in *Figure 6 (A,B)*; originating gels in *Figure 3—figure supplement 1*. The antibody, raised against the 130 kDa α1 subunit of this protein, consistently stained two bands, one at the expected 130 kDa and another at between 110 and 115 kDa. The densities of the two bands are positively correlated (*Figure 6—figure supplement 1*), which suggests that the 115 kDa band contains a large fragment of the subunit. The distribution of the 2-band average is plotted in panel 6E. The mean signal in microsomes was almost 100% higher in the MHS (p = 4 $10^{-4}$). In the whole muscle fraction, by contrast, the mean signal increased slightly and not significantly (*Table 1*).

The rate of glycogen synthesis in muscle is determined by the activity of glycogen synthase (GS). GS and GP are phosphorylated by PhK (*Krebs et al., 1959*). If the increase in GP*a* were due to the greater activity of PhK, it would be associated with an increase in GS*a*. The ratio GS*a*/GS was

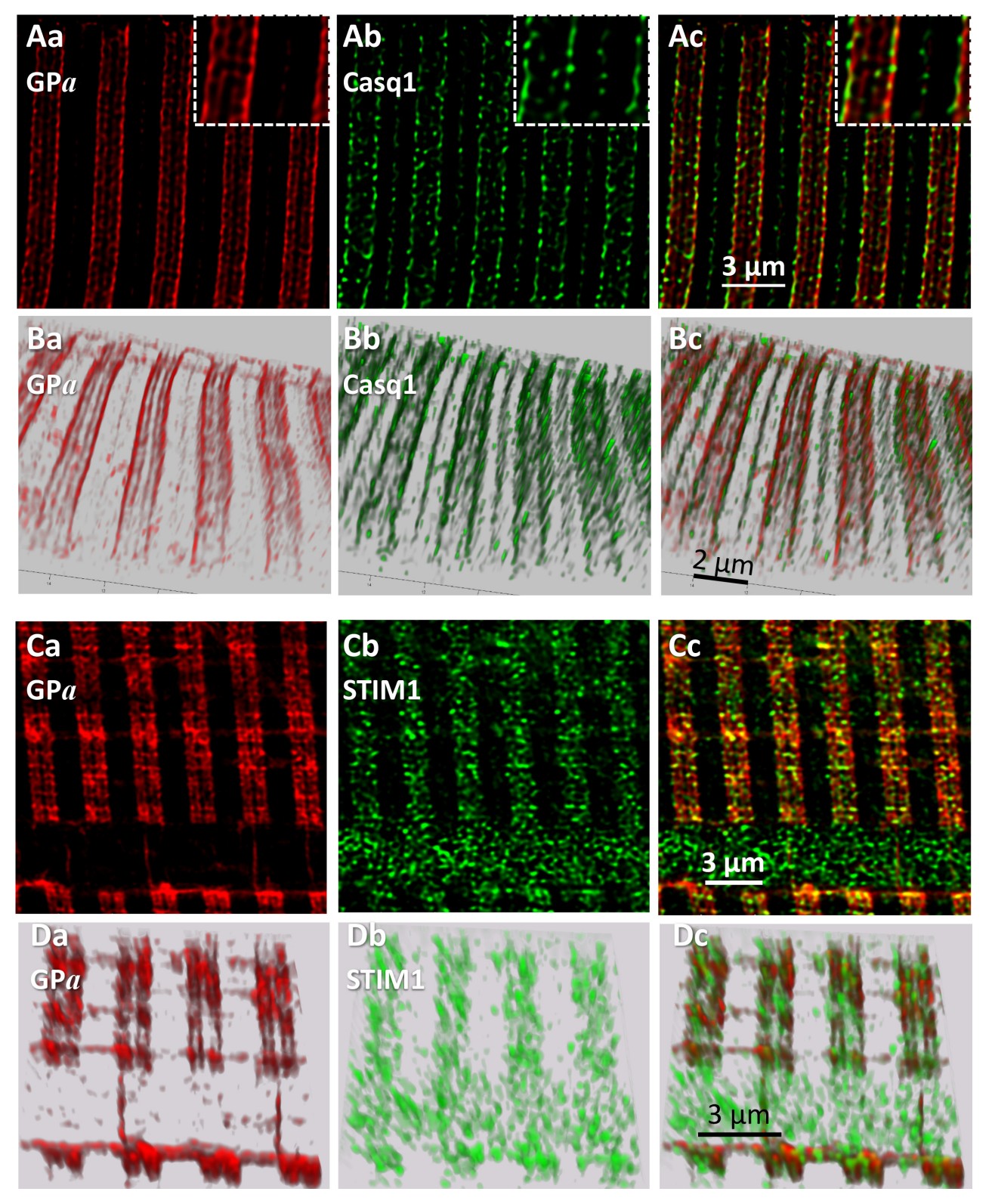

**Figure 5.** Phosphorylated glycogen phosphorylase, GP*a*, localizes at SR. (**A**) Sections of a z-stack (corrected as in *Figure 4*) of human muscle co-stained for GP*a* and Casq1(I.D. #164, MHN). (**B**) 3-D renderings of the z-stacks that contained the sections in A. Both proteins delineate two terminal cisternae, TC, in every triad. The inset in Ac shows close proximity of the imaged proteins, without actual colocalization (documented with *Figure 5—figure supplement 1*). (**C, D**) Sections and stack from a human myofiber co-stained for GP*a* and STIM1 (I.D. #164, MHN). GP*a* delineates two TC in

*Figure 5 continued on next page*

Figure 5 continued

every triad (Ca, Da). STIM1 forms clusters in close proximity to the GP*a* bands, without actual colocalization. Minor quantities of both proteins are located outside triads. A large quantity of STIM1, unaccompanied by GP*a*, appears inside a nucleus (cf. (*Lee et al., 2018*). These images are the first to resolve TC with conventional optics (a resolution of ~100 nm is estimated with *Figure 5—figure supplement 2*).

The online version of this article includes the following figure supplement(s) for figure 5:

**Figure supplement 1.** Analysis of colocalization of GP*a* and STIM1.

**Figure supplement 2.** Interpretation of high resolution GP*a* images.

---

determined in the microsomal fraction of all patients that had PhK evaluated. The GS blots, in panels 6C and D, consistently show two bands. The lighter band might correspond to a GS fragment or to the GS liver isoform. Here they are identified as bands 1 and 2. The median ratio GS*a*/GS for band one was higher by 80% in MHS patients (*Figure 6G*) but the difference was not significant (p = 0.11, *Table 1*). Band two was not substantially different.

Although the increase of GS*a*/GS in MHS cells was not significant by conventional criteria, it was significantly positively correlated with the PhK content in microsomes, as well as with the level of phosphorylation of GP in whole muscle (Panels 5K, L, and *Table 2*). The correlations reduce the likelihood that the changes be due to random variability, revealing instead an ample remodeling of glucose metabolic pathways.

## Large increases in phosphorylation can be induced by changing cytosolic calcium

Put together, the present observations support a specific causal chain for the observed changes in metabolic enzymes, leading to a reduction in glycogen content and eventually hyperglycemia. To place this chain within the pathophysiological context of our patients, we probed a possible link between the MHS condition and the observed changes in GP phosphorylation.

The conversion of GP*b* to GP*a* by PhK (the original 'converting enzyme'; *Krebs and Fischer, 1956*), was the first phosphorylation process known to cause functional changes in the substrate. The regulation of PhK is complex (rev. *Brushia and Walsh, 1999*). $Ca^{2+}$ promotes its activation by direct allosteric action and indirectly by enhancing its phosphorylation by PKA. Myofibers from animal models, patients with MHS and myotubes developed from MHS patients' muscle have resting $[Ca^{2+}]_{cyto}$ elevated to between 150 and 500 nM (*López Padrino, 1994*; *Lopez et al., 1986*; *Figueroa et al., 2019*). We tested whether changes in this range evoke an increase in phosphorylation. Biopsied muscle from four individuals, 3 MHN and 1 MHS, provided sufficient material for a stringent test. Two groups of 10 bundles of 10–40 fibers dissected from the same biopsy were pinned slightly stretched in two Sylgard-bottom chambers, exposed to saponin for permeabilization and then to two $[Ca^{2+}]$, 100 and 500 nM. One of the biopsies provided enough tissue for an additional $[Ca^{2+}]$, 300 nM. After 10 min, the muscles exposed to different $[Ca^{2+}]$ were processed separately to whole tissue extracts. As many extracts as applied $Ca^{2+}$ concentrations were produced for every patient and were aliquoted for quantitative blotting of GP and GP*a* in separate gels. Blots and gels are shown in *Figure 7*. The ratio of signals for GP*a* and GP evolved upon change in $[Ca^{2+}]$ as plotted in *Figure 7C*, always increasing at the higher $[Ca^{2+}]$ (p < 0.001 in a *t* test of paired differences).

## Glycogen and GLUT4 levels are reduced in the membrane fraction of MHS muscles

Correlated increases in phosphorylation of GP and GS suggest a concerted control process that, by promoting activation of GP and inactivation of GS, tilts the glucose ⟵⟶ glycogen balance towards glycogenolysis and away from glucose storage. In mammalian and especially in human muscle, glycogen is found largely in intermyofibrillar regions (*Ørtenblad and Nielsen, 2015*; *Entman et al., 1980*; *Cuenda et al., 1994*; *Lees et al., 2001*), and partitions with the microsomal fraction in biochemical analyses. The Glycogen content of microsomes was compared in the 13 MHN and 12 MHS patients studied for glycogen processing enzymes. (A test that showed minor not significant differences between samples shipped at 4°C and others kept frozen is described in Materials and methods). The results, presented in *Figure 8D* and *Table 1*, were expected from the enzymatic changes: the

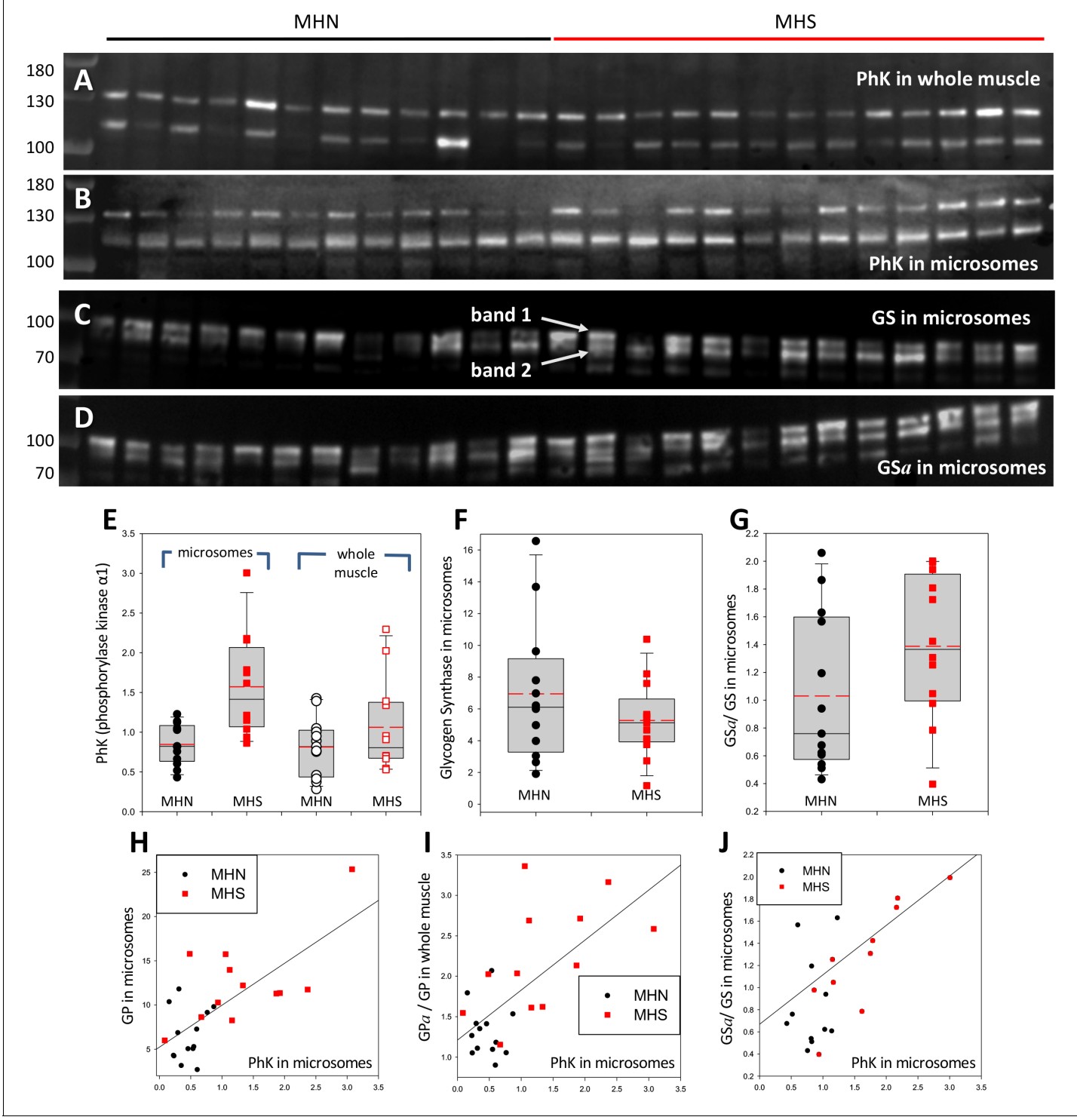

**Figure 6.** Phosphorylase kinase and glycogen synthase are enriched in MHS microsomes. (**A, B**) Blots stained for PhK in whole muscle (**A**) or microsomes (**B**) from the patients identified in *Figure 2*. Two bands are stained. The correlation of their intensities (in *Figure 6—figure supplement 1*) indicates that the 115 band contains a lysis product of the ~130 kDa protein. (**C, D**) Blots of *all-forms* glycogen synthase (GS, consisting of GS*a* and GS*b*) and the phosphorylated enzyme (GS*a*). Originating electrophoretic gels for A-D are in *Figure 3—figure supplement 1*. Two bands, named 1 and 2, are stained. (**E–F**) The content of the PhK bands from microsomes is nearly double in the MHS patients (p = 5 $10^{-4}$). In whole muscle the difference is not significant. Differences in the content of GS band one are small and not significant, but the median phosphorylation (GS*a*/GS) almost doubles (p = 0.11). GS band two does not change significantly (not shown). (**H–J**) The graphs show positive correlations between PhK content, phosphorylation

*Figure 6 continued on next page*

*Figure 6 continued*

of GP and GS and content of GP in microsomes. The correlation coefficients are high and *p*s of no correlation are uniformly low (*Table 2*). Remarkably, the microsomal PhK content correlates significantly with the degree of phosphorylation of GP in whole muscle, not in microsomes.

The online version of this article includes the following figure supplement(s) for figure 6:

**Figure supplement 1.** Analysis of two bands stained by the anti-PhK antibody.

median glycogen content was reduced in the microsomes of MHS patients by 43% (p = 0.01). Glycogen was negatively correlated with PhK and GP*a* contents of the microsomal and whole muscle fractions, as illustrated for whole muscle GP*a* in *Figure 8B* (r = -0.47, p = 0.02). Glycogen was also negatively correlated with the microsomal content of GDE and phosphorylated GS, but with lower significance (parameters listed in *Table 2*). The uniformly negative correlations suggest that the changes in glycogen are driven by the reciprocal changes in activity of GP and GS due to the increased activity of their kinase.

Together with the observation of a greater incidence of hyperglycemia in MHS, *Altamirano et al., 2019* found a deficit in phosphorylation of Akt, a kinase that mediates the muscle response to insulin. This response involves recruitment of the glucose transporter GLUT4 to the plasma membrane (*Beg et al., 2017*). A comparison of the GLUT4 content of the microsomal fraction in MHN and MHS patients is documented in *Figure 8C–E*. The content was lower in the MHS by more than 60% (p = 0.01, *Table 1*). There were negative correlations between GLUT4 and PhK and GP*a* contents of the microsomal and whole muscle fractions, illustrated for whole muscle GP*a* in *Figure 8E* (r = -0.46, p = 0.02). This result is consistent with the proposed association of the MHS

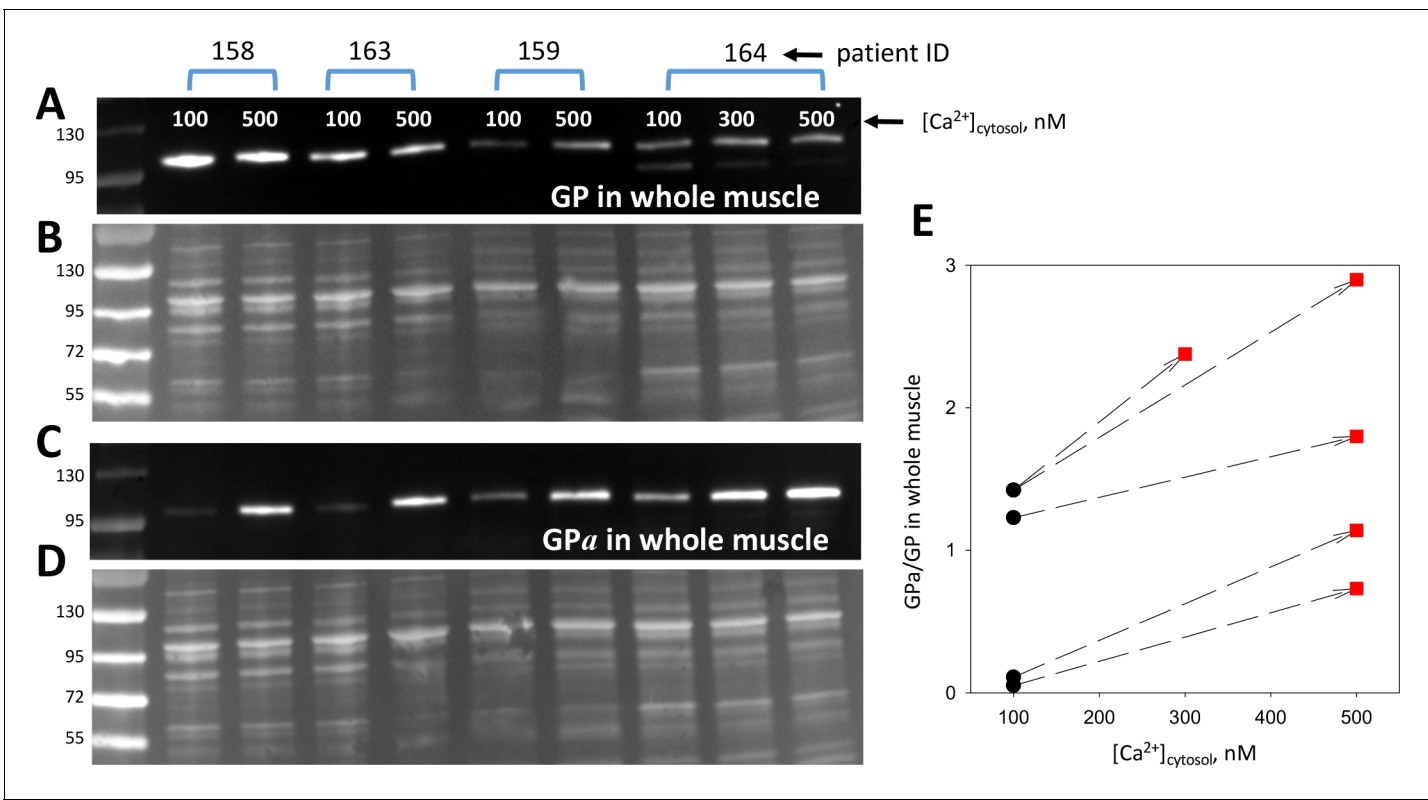

**Figure 7.** Phosphorylation is modulated by cytosolic calcium. (**A, C**) Blots of GP and GP*a* in whole tissue fractions of permeabilized muscle bundles from four patients, exposed successively to two solutions of increasing [Ca$^{2+}$]. (**B, D**) Protein gels from which blots in A and C were derived. Corresponding lanes in the two gels were loaded with the same extract, prepared from multiple, equally treated fiber bundles from the patients identified at top. (**E**) Applied [Ca$^{2+}$]$_{cyto}$ vs. phosphorylation ratio, calculated from band signals in A and C after normalization by protein content in B and D. Phosphorylation increased at 500 nM [Ca$^{2+}$] (p = 8 10$^{-4}$). The effect was graded with [Ca$^{2+}$] in the sample from patient # 164, which provided sufficient tissue for applying three Ca$^{2+}$ concentrations.

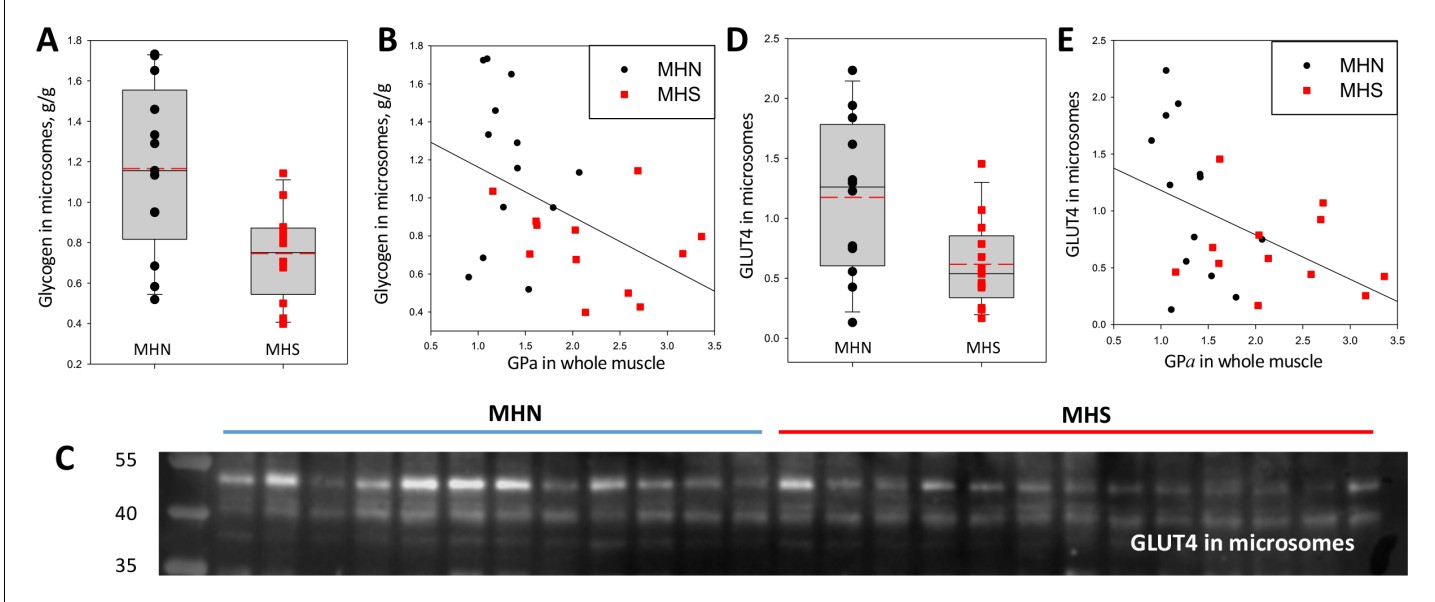

**Figure 8.** Glycogen and GLUT4 are lower in MHS patients. (A) Distribution of glycogen content in microsomes. The median was 43% lower in MHS (p = 0.01). (B) Negative correlation of glycogen in microsomes with GPa in whole tissue (r = - 0.47, p = 0.02). (C) Western blot of GLUT4 in microsomes. The protein of interest is the band at ~50 kDa. (D) Distribution of signal intensity; the median content was lower by 54% in the MHS (p = 0.01, *Table 1*). (E) Negative correlation of GLUT4 in microsomes with GPa in whole tissue (r = - 0.46, p = 0.02). The gel originating the blot in C is panel F in *Figure 3—figure supplement 1*.

condition with increased insulin resistance (*Altamirano et al., 2019*), mediated by a reduced availability of GLUT4.

## Biochemical analyses of an alternative group of patients

Samples from a different group of patients were subjected earlier to a more limited study of its proteins. The results are described in Appendix 2; they are consistent with those presented in the main text.

## Discussion

Our study reveals that Malignant Hyperthermia Susceptibility entails a wide-ranging, pathogenic change in glucose metabolism. The present quantification of glucose processing enzymes in muscle of human subjects, combined with high resolution imaging of tagged proteins, demonstrated altered amounts of many of these molecules in patients susceptible to Malignant Hyperthermia. Multiple features of these changes indicate that they result from the primary defect in $Ca^{2+}$ management that underlies the MHS condition.

Susceptibility to Malignant Hyperthermia, diagnosed by the CHCT, is an inheritable condition with variable disease phenotype and multiple causes. Some susceptible individuals have pathogenic mutations in the couplon proteins, including RyR1, the $Ca^{2+}$ release channel of the SR (*Rosenberg et al., 2015*; *Robinson et al., 2006*), its voltage sensor $Ca_V1.1$ (*Robinson et al., 1997*; *Carpenter et al., 2009*) and the 'connector' protein STAC3 (*Webb et al., 1993*; *Horstick et al., 2013*). Other MHS patients do not have a known diagnostic mutation, but the commonality in clinical features and cellular alterations in MHS animal models (*Lopez et al., 1986*; *MacLennan and Zvaritch, 2011*) and human cells (*Figueroa et al., 2019*) indicates that all have a primary defect originated in the couplon (*Stern et al., 1997*; *Franzini-Armstrong et al., 1999*), generally causing excessive release of $Ca^{2+}$ from the SR. The consequence is partial depletion of SR calcium, which activates compensatory calcium entry into the cell (SOCE; *Michelucci et al., 2018*) resulting in elevation of $[Ca^{2+}]_{cyto}$. The present studies sought mechanistic explanations for the high incidence of hyperglycemia found in MHS subjects (*Altamirano et al., 2019*).

Starting from changes visible even without specific staining in gels of the microsomal fraction, the main observations will be summarized with references to *Tables 1* and *2*. As listed in *Table 1*, where significant changes are coded in color, the MHS samples had substantial increases — in microsomes and whole tissue lysates — of GP, its phosphorylated form GP*a*, the phosphorylated fraction GP*a*/GP, phosphorylation of glycogen synthase (GS*a*/GS), and content of the GP- and GS-kinase PhK. As expected from these changes, microsomal glycogen was substantially reduced. Additionally, we found less GLUT4 in microsomes, a change expected in view of the reduced phosphorylation of Akt in response to insulin in MHS mouse myotubes described by *Altamirano et al., 2019*.

Patients who had a marked increase in PhK were found to have substantial changes in the other variables as well. This convergence, quantified by high correlation coefficients between variables (*Table 2*), indicates that the changes in PhK content cause the increases in phosphorylation of GP and GS and the decrease in glycogen content. The correlations bolster the statistical significance of the changes — if patients who have high GP*a* also have high PhK or low glycogen, it is unlikely that the high values be due to random variability or measurement error. Multivariate analysis documented in Appendix 3, of correlations between more than two variables, strengthen the conclusion that the changes in GP, GP*a* and glycogen content are caused by activation of PhK. These studies also attribute high significance to the changes in content of GDE and GLUT4. Elevated GP*a* in MHS was first reported in *Willner et al., 1980*, but later questioned due to the large number of MHN patients with high GP*a* (*Traynor et al., 1983*). While we also found false positives among our patients, there was a clear association between the MHS condition and a wide-ranging change in glucose processing enzymes, which included the excess GP*a*. (A different validation, based on the estimation of probable error, is presented in the section Materials and methods/Statistics/Replicates and Probable Error).

In sum, the results demonstrate in MHS individuals an activation of the pathway of glycogen breakdown to glucose, which shares many features with the program activated during exercise, including a dependence on $[Ca^{2+}]_{cyto}$, parallel phosphorylation of GP and GS, and decrease in glycogen (rev. *Mul et al., 2015*). However, the changes in MHS muscle differ by including an increase in GDE and a decrease in GLUT4 in microsomes. The changes in GDE and GLUT4 complete a wide-ranging metabolic alteration that promotes glycogen breakdown and may hamper glucose uptake and utilization inside muscle, thus promoting hyperglycemia.

The observed decrease in GLUT4 in microsomes is not immediately relatable to the steady increase in $[Ca^{2+}]_{cyto}$ that we propose as primary mediator of the metabolic changes in MHS patients. On the contrary, the increase in $[Ca^{2+}]_{cyto}$ that mediates EC coupling (*Valant and Erlij, 1983*; *Youn et al., 1991*), plus that caused by $Ca^{2+}$ entry in response to insulin (*Lanner et al., 2009*) are believed to promote GLUT4 expression and its translocation to the plasma and transverse tubular membranes. The physiological increase in GLUT4 activity in response to insulin is preceded by activation of phosphatidyl-inositol 3-kinase, subsequent recruitment of Akt to the plasma membrane and its phosphorylation by 3-PI-dependent protein kinase 1 (PDK1). Akt phosphorylation follows $Ca^{2+}$ entry, in muscle and also in cultured neurons, where it is associated with electrical activity (*Nicholson-Fish et al., 2016*) and in osteogenic cells, where $Ca^{2+}$ entry follows mechanical stimulation (*Danciu et al., 2003*). These observations favor a positive effect of cytosolic $Ca^{2+}$ on GLUT4 activity, by increasing its expression or deployment to the cell boundaries.

On the other hand, *Park et al., 2009* demonstrated that GLUT4 expression is inhibited by chronic elevation of $[Ca^{2+}]_{cyto}$, in agreement with negative effects of prolonged exposure of muscle to ionomycin (*Lee et al., 1995*). These interventions specifically abolished the activation of GLUT4 expression by AMP-activated protein kinase (AMPK), an insulin-independent effect mediated by $[Ca^{2+}]_{cyto}$ (*Ihlemann et al., 1999*). The time course of $[Ca^{2+}]_{cyto}$ determined the sign of the effect: promotion of GLUT4 required $Ca^{2+}$ transients, while a sustained increase in the ion concentration had the opposite effect. In conclusion, the chronic increase in $[Ca^{2+}]_{cyto}$ that occurs in MHS could justify, via distinct but entangled mechanisms, both the increased phosphorylation of PhK, GP and GS and the reduced content of GLUT4 observed in these patients. In this view, the reduction in glycogen content would be caused by both the imbalance in the pathway to and from glucose, and a reduction in uptake and availability of glucose. The main elements of this interpretation are diagrammed in *Figure 9*.

Because the alterations in MHS patients include reduction in the main insulin-dependent glucose transporter, the condition may lead to insulin resistance. Elevated glycogen breakdown and lower

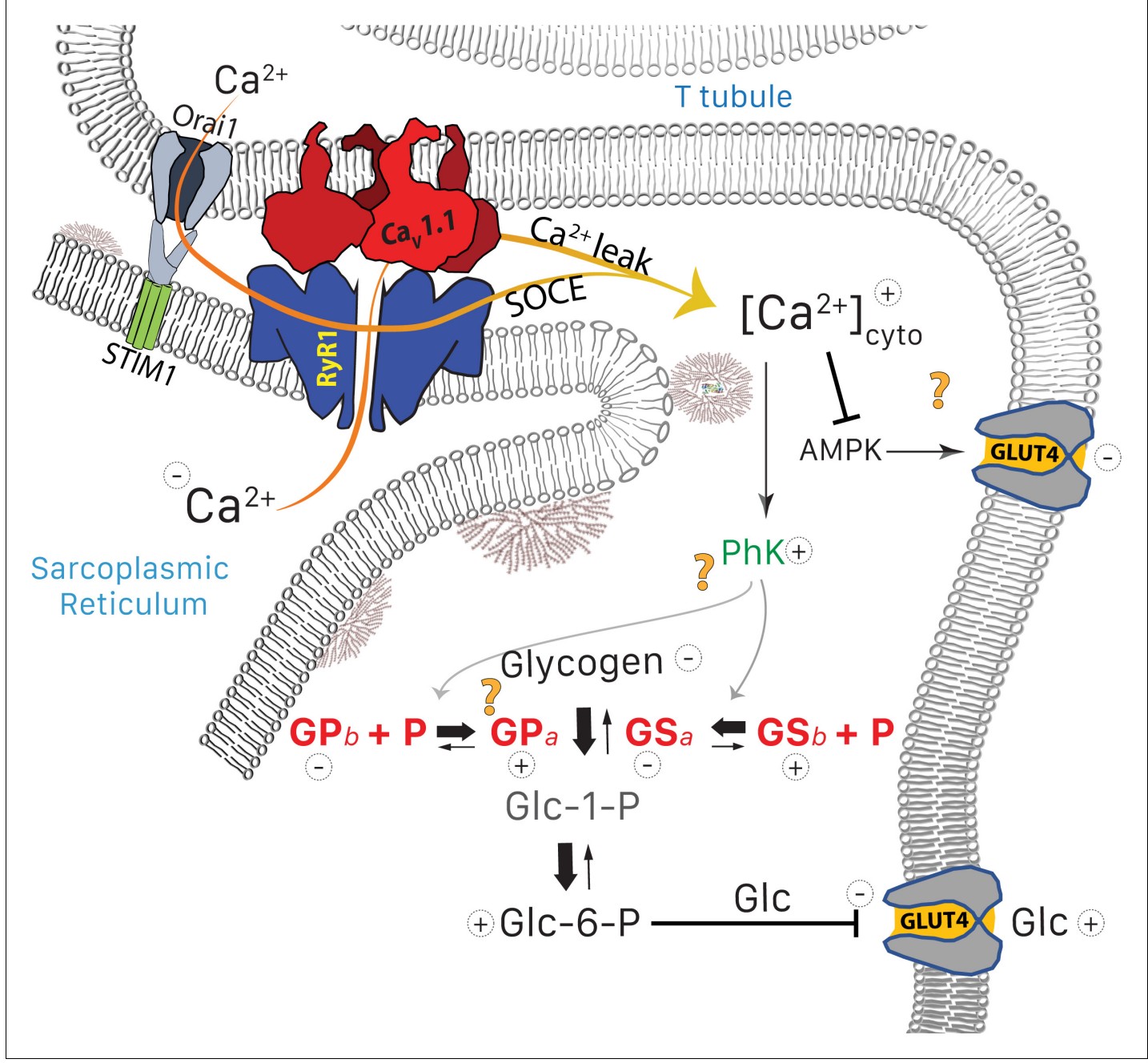

**Figure 9.** Glucose pathways and their alterations in MHS muscle. Glycogen, resident in the intermyofibrillar space in the form of granules or SR-bound, is synthesized from transported glucose or broken down to glucose-1-phosphate (Glc-1-P). In MHS, $[Ca^{2+}]_{cyto}$ is elevated (*López Padrino, 1994*) as a consequence (*Rios, 2012*) of an increased leak from the SR that results in a lower resting free SR calcium level. The lower $[Ca^{2+}]_{SR}$ promotes influx via SOCE or other pathways (*Eltit et al., 2013*) resulting in higher $[Ca^{2+}]_{cyto}$. The change has two effects: (i) PhK activation that increases phosphorylation of GP and GS, to respectively enhance the breakdown and decrease the synthesis of glycogen. The resulting increase in Glc-1-P, Glc-6-P and glucose (Glc) hampers passive entry of glucose. (ii) Decrease of the basal activation by AMPK of the expression and membrane insertion of the glucose transporter GLUT4 (61), which contributes to reducing glucose entry. Concurrently, PhK, GPa, GSa and GDE move towards the intermyofibrillar space (not shown). Question marks indicate stages where evidence is lacking or conflicting. They include the site of interaction of PhK with its substrates, the location of GPa (whether in granules or attached to TC) and the possible effects of elevated $[Ca^{2+}]$ on GLUT4.

flux of glucose into storage inside muscle suggest that the cytosolic concentration of glucose would increase, which would lower the gradient and further impair the response to insulin in MHS subjects.

The metabolic alteration in muscle metabolism is associated with hyperglycemia. This is demonstrated by the relationship between FBS and measures of phosphorylation in the present study. As shown in *Figure 10*, a positive correlation exists between FBS and the ratio GP*a*/GP of whole tissue ($r$ = 0.58), which implies that the metabolic alteration detected among the MHS has systemic consequences that favor high FBS. A rough quantitative measure of the relevance of the metabolic change as a determinant of hyperglycemia was derived from the slope ($b$ = 2.5 mM) of the linear fit in the figure. In the sample studied, the abscissa (GP*a*/GP) increased by 0.15, from a median value of 0.36 in the MHN to 0.51 in the MHS (Table I). An increment of 0.15 in the abscissa corresponds on the regression line to a difference in FBS of 2.5 mM ×0.15, or 0.38 mM higher sugar. The analysis of 'legacy' patients (illustrated with *Figure 1*) showed an excess FBS averaging ~0.5 mM in MHS at all ages. Extrapolation of the present study to the 'legacy' cohort suggests that a major part of the excess FBS (0.38/0.5 mM) is related to and potentially explained by the measured changes in glycogen processing enzymes. The remainder could be explained by other alterations that increase blood sugar, not defined, manifested in the separate correlations of hyperglycemia with age and BMI demonstrated in the 'legacy' cohort (*Figure 1*). Jointly with the observations of *Altamirano et al., 2019*, the present results identify MHS as a prodrome and cause of hyperglycemia. The alterations justify the increase of the incidence of type 2 diabetes observed in the MHS sample group (22%) over that in the age-matched Canadian population (10%).

In view of this analysis, it was surprising that neither the difference between FBS averaged over all MHS and MHN patients of the 'recent' cohort (*Figure 1—figure supplement 1*) nor that between the 13 MHN and 12 MHS patients fully studied (*Figure 10* and row 18 in *Table 1*) reached statistical significance. A much greater difference was found between groups defined by metabolic variables, namely the MHS patients that had the highest content of PhK and GP in microsomes and patients with low content of these proteins. While the selection was based on PhK and GP only, the top six also had the highest GP*a* content and GP*a*/GP ratio in whole muscle lysate, the highest GDE and GS*a*/GS, and the lowest GLUT4 in microsomes; we call them *MC*, for metabolically challenged. *MC* had an average FBS of 5.90 mM (*Table 1*, row 19). The metabolically normal or *MN* (six patients with low values of PhK and GP, all of whom were *MHN*) had FBS = 5.05 mM (*Table 1*). The difference between *MC* and *MN* was highly significant. The weaker association found between FBS and F$_H$ (the

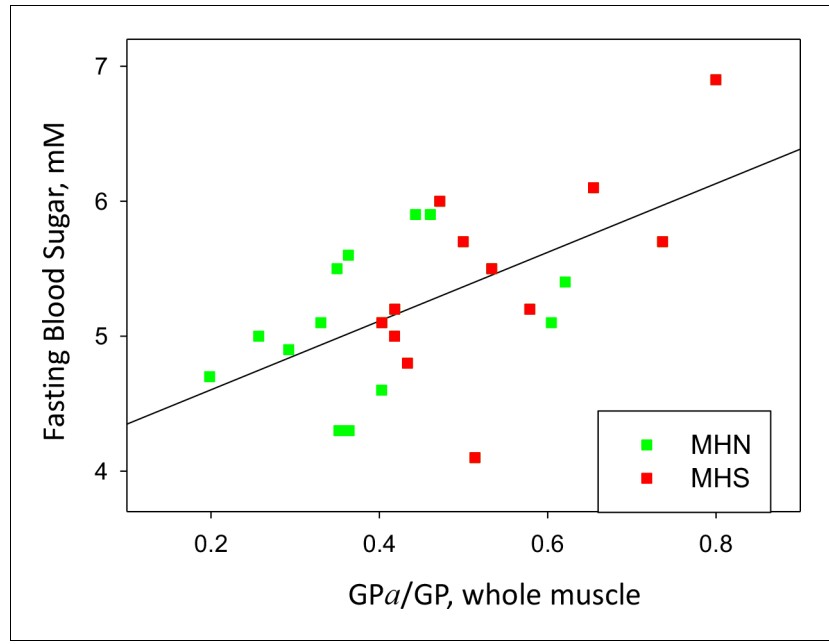

**Figure 10.** GP phosphorylation and glycemia. Phosphorylation ratio of glycogen phosphorylase in the whole tissue extract, vs. FBS, for the 25 patients with a complete study. $r$ = 0.58. p=0.002.

measure on which MHS diagnosis is based) might reflect links between calcium homeostasis and glycogen metabolism that remain unknown and uncorrelated with $F_H$, or just the challenging nature of the CHCT.

As a second outcome, the present work promotes the use of samples from MHS subjects as 'tools' to explore normal physiology, specifically harboring a perturbation that stresses otherwise normal pathways. Note first that MHN and MHS patients are classified based on a threshold value of the halothane-induced force $F_H$. $F_H$ has a unimodal distribution among individuals (inset in *Figure 1A*), with probability density varying monotonically near the MHS threshold. This implies that there is no unique way of setting this threshold; the classification is conventional rather than clearcut.

Because the MHS-defining variable $F_H$ is graded, the associated deviations from normalcy in metabolic properties should also be graded. In agreement, none of the variables measured showed any clustering that would define a boundary between normalcy and disease. An absence of clustering was also the case in the cell-level study of calcium signals of the 'recent' cohort (*Figueroa et al., 2019*). Within this context of graded variation, the quantitative correlations linking changes in the measured variables suggest that MH susceptibility alters continuously a cellular parameter critical for the processing of glucose towards glycogen. The graph in *Figure 2H* provides an example: the good correlation between the quantity of GP and GDE in microsomes extends smoothly from MHN to MHS patients, without obvious change –in regression coefficient or other properties—between the two groups. The MHS condition — presumably through its higher $[Ca^{2+}]_{cyto}$ — appears to impose changes, such as forcing more of the glycogenolytic enzymes towards the SR, without changing essential properties.

The stress provided by the MHS condition affords multiple insights into normal pathways: first it reveals a close relationship between abundance of PhK and phosphorylation of GP, which in turn leads to migration of the enzyme to the microsomal fraction. Imaging at unprecedented resolution locates the phosphorylated GP at or near the SR, predominantly at terminal cisternae. An association between GP and the SR has been reported since 1972 (*Wanson and Drochmans, 1972*) but the effect of GP phosphorylation on this linkage remained obscure. We now find that the content of GP in microsomes is strictly correlated with its degree of phosphorylation in the whole tissue extract, but not with that in microsomes. This is possible if all GP in microsomes is in the GP*a* form. This inference, derived from biochemical data in lysates (*Figure 3*), is consistent with the distribution of GP*a* imaged in muscle biopsies (*Figure 5*) and of GP*b* derived tentatively by linearly combining GP and GP*a* images (*Figure 4Dc*, E). The present evidence contradicts previous reports that found mostly GP*b* in a rabbit SR fraction, the phosphorylation of which caused its detachment from the SR (*Cuenda et al., 1994*; *Cuenda et al., 1995*).

Glycogen granules 10–40 nm in diameter (*Prats et al., 2018*) share the intermyofibrillar space with the SR (*Ørtenblad and Nielsen, 2015*; *Fridén et al., 1989*) and are part of the microsomal fraction. They contain multiple proteins (including GP, GDE, GS, PhK, PP1, PKA and glycogen branching enzyme). While GP detected in the microsomal fraction might in principle reside in its glycogen granules, the irregular distribution of the granules in the inter-myofibrillar space (e.g. *Ørtenblad and Nielsen, 2015*) is inconsistent with the detailed delineation of terminal cisternae by the specific GP*a* antibody (*Figure 5C,D* and its supplements). The present images suggest instead a detailed and extensive binding of GP*a* to the SR, perhaps mediated by glycogen as proposed in early studies of GP that distinguished direct and granule-mediated association (*Wanson and Drochmans, 1972*; *Hirata et al., 2003*).

The changes in MHS include enrichment of GDE in the microsomal fraction. GDE, however, diminishes together with glycogen in microsomes of exercised rats (*Lees et al., 2004*). Again, the observations can be reconciled assuming that GDE has means to associate with the SR other than via glycogen granules.

While PhK promotes phosphorylation of GP and GS (*Brushia and Walsh, 1999*; *Walsh et al., 1979*), the changes in GS observed here differ from those of GP: GS*a*/GS increases in the microsomal fraction while its total content does not (*Table 1*). The observations imply that GS*a* and GS*b* coexist in the membrane system and their relative amounts are regulated in response to changes in $[Ca^{2+}]$.

As already noted, PhK increases in the microsomes but remains constant in the whole tissue extract. The change thus appears to be a migration of the kinase towards the membrane fraction.

The functional implications of PhK migration are intriguing, considering that GP*b* — its substrate — remains at the I band, near the Z disk (*Figure 4* and *Maruyama et al., 1985*; *Chowrashi et al., 2002*), while GP*a* appears near the SR. The strong heterogeneity in distribution of PhK and GP*a* might be related to the standing gradient of $[Ca^{2+}]$ between triadic space and bulk cytosol proposed to emerge from the operation of SOCE channels preassembled at T-SR junctions (*Cully et al., 2018*). This conundrum and other questions about mechanism left unanswered by the present study, are stressed by question marks in the functional diagram of *Figure 9*.

The present findings have implications for clinical management. Follow-up of the six patients identified here as metabolically challenged encountered a disease phenotype, quantified by the Clinical Index defined in *Figueroa et al., 2019*, greater than average for the MHS class. The observation suggests that the alterations in glycogen metabolism cause additional health impairment. GP Inhibitors decrease FBS (*Nagy, 2017*; *Docsa et al., 2011*; *Pałasz et al., 2019*) and increase insulin sensitivity (*Docsa et al., 2015*). The finding of tight links between calcium signaling and glycogen metabolism suggests the use of GP inhibitors not just to prevent hyperglycemia, but also to treat the MHS condition. Conversely, treating the causes of abnormal $Ca^{2+}$ handling in MHS should moderate the changes in glycogen metabolism, as already reported by *Altamirano et al., 2019* in their study of an MHS animal model.

The present observations prescribe follow-up of all MHS subjects, watchful for the development of diabetes. In animal models, the presence of GP in extracellular circulating vesicles is an early marker of cardiac injury (*Yarana et al., 2018*); in MHS, the increase of GP*a* and other proteins precedes hyperglycemia, therefore the potential of circulating glucose metabolism enzymes as its early markers should be evaluated.

The present study has limitations: the resting $[Ca^{2+}]_{cyto}$ in muscle —putative proximate cause of the metabolic alterations — is not known for these individuals. As an approximation, $[Ca^{2+}]_{cyto}$ is being measured in myotubes derived from biopsied muscle (a systematic elevation was found in cells from MHS patients; *Figueroa et al., 2019*). To what extent these alterations result in diabetes will be assessed by the response of MHS and MHN myocytes to insulin and glucose tolerance tests on patients.

The imaging of GP and GP*a* presented here will serve as blueprint for studies of other molecules, including PhK — to reveal where phosphorylation of its targets occurs — and mediators of the insulin response in healthy and diseased individuals. The visualization of exogenous GP in living myocytes (*Figure 4C*) demonstrates the feasibility of a different approach: imaging the movements of glucose regulatory proteins during normal function, under imposed stress and upon interventions of potential therapeutic value.

## Materials and methods

**Key resources table**

| Reagent type (species) or resource | Designation | Source or reference | Identifiers | Additional information |
|---|---|---|---|---|
| Antibody | anti-PYGM (Rabbit Monoclonal) | Invitrogen | Catalog #MA5-27442 | IF(1:200), WB (1:2000) |
| Antibody | anti-AGL (Rabbit Polyclonal) | Thermo Fisher scientific | Catalog # PA5-12142 | WB (1:1000) |
| Antibody | anti-PhKA1 (Rabbit Polyclonal) | Thermo Fisher scientific | Catalog #PA5-51508 | WB (1:1000) |
| Antibody | anti-glycogen synthase (Rabbit Polyclonal) | Cell Signalling Technology | Catalog #3893S | WB (1:1000) |
| Antibody | anti- phosphorylated glycogen synthase (Rabbit Polyclonal) | Cell Signalling Technology | Catalog #3891S | WB (1:1000) |
| Antibody | anti-phosphorylated glycogen phosphorylase (Sheep polyclonal) | MRC PPU Reagents | Clone #cS960A | IF(1:200), WB (1:2000) |

*Continued on next page*

*Continued*

| Reagent type (species) or resource | Designation | Source or reference | Identifiers | Additional information |
|---|---|---|---|---|
| Antibody | anti-SERCA1 (Mouse Monoclonal) | Thermo Fisher scientific | Catalog #MA3-912 | WB (1:2000) |
| Antibody | anti-FKBP12 (Rabbit Polyclonal) | Thermo Fisher scientific | Catalog # PA1-026A | WB (1:1000) |
| Antibody | anti-Casq1 (Mouse Monoclonal) | Thermo Fisher scientific | Catalog #MA3-913 | IF(1:200), WB (1:1000) |
| Antibody | anti-Glut4 (Mouse Monoclonal) | Santa Cruz Biotechnology | Catalog #Sc-53566 | WB (1:500) |
| Antibody | Anti-Actinin (Mouse Monoclonal) | Millipore sigma | catalog #A7811 | IF(1:200) |
| Antibody | anti-Stim1 (Mouse Monoclonal) | Invitrogen | catalog # MA1-19451 | IF(1:200) |
| Antibody | Anti-cox4 (Mouse Monoclonal) | Thermo Fisher scientific | Catalog #MA5-15686 | WB (1:1000) |
| commercial assay or kit | Glycogen assay | Sigma | Catalog #MAK016 | |
| recombinant DNA reagent | Pygm-GFP (Plasmid) | Sino Biologicals | Catalog # MG52459-ACG | |
| Biological sample (*Homo-sapiens*) | *Gracilis* muscle | MHIU, Toronto Gen. Hospital | unique patient # and date of collection | |
| Biological sample (*Mus musculus*) | *FDB* muscle | Charles River Laboratories, Boston, MA | Black Swiss | |

## Patients

Criteria for recruitment of subjects included one or more of the following: a previous adverse anesthetic reaction, family history of MH without a diagnostic MH mutation (www.emhg.org), a variant of unknown significance (VUS) in *RYR1* or *CACNA1S*, recurrent exercise- or heat-induced rhabdomyolysis and idiopathic elevation of serum creatine kinase.

We distinguish two groups: a 'legacy' cohort of 560 individuals diagnosed at the Malignant Hyperthermia Investigation Unit (MHIU) of Toronto General Hospital (TGH) in the years 1994 to 2013, who underwent clinical studies and CHCT, and a 'recent' cohort of individuals diagnosed since 2014, which were studied jointly at the MHIU and the Rush University laboratory. The studies communicated here include the analysis of 11 protein species and glycogen in microsomal and whole muscle fractions of 25 patients of the 'recent' cohort, 12 MHS and 13 MHN, chosen because the muscle specimens had a regular striation pattern of > 2 μm per sarcomere consistent with a relaxed state and provided sufficient tissue for biochemistry and imaging. An alternate group of 26 samples, 14 MHN and 12 MHS, underwent a less complete study, yielding similar results summarized in Appendix 2. All specimens from patients of the recent cohort were de-identified and assigned a unique number before arrival to the Rush laboratory; their identity was known only to the attending personnel at the MHIU.

## Mice

6–10 wk-old mice, *Mus musculus*, of the Black Swiss strain, sourced at Charles River Laboratories (Boston MA, USA), were used to define the localization of GP in living cells. Hind paws were transfected with plasmid vector for GP-GFP as described in *Pouvreau et al., 2007*. Animals were euthanized and muscles collected and processed for imaging as in *Manno et al., 2017*.

## CHCT

Susceptibility to MH was diagnosed following the North American CHCT protocol (*Larach, 1989*). Increases in baseline force in response to caffeine and halothane ($F_C$ and $F_H$) were measured on freshly excised biopsies of *Gracilis* muscle with initial twitch responses that met viability criteria.

Three muscle bundles were exposed successively to 0.5, 1, 2, 4, 8 and 32 mM caffeine; three separate bundles were exposed to 3% halothane. The threshold response for a positive diagnosis was either $F_{Hx00A0} \geq 0.7$ g or $F_C$ (2 mM caffeine)$\geq 0.3$ g. Patients were diagnosed as 'MH-negative' (MHN) if the increase in force was below threshold for both agonists, and 'MH-susceptible' (MHS) if at least one exposure exceeded the threshold.

## Protein fractionation, Western blotting and quantitative analysis of Western blots

Biopsied segments were shipped from the MHIU to Rush University (Chicago, USA) at 4°C overnight in relaxing solution (*Figueroa et al., 2019*). Upon receipt, thin bundles were dissected for immunohistochemistry or physiological study, small pieces were separated for generation of cell cultures and the remaining tissue was quick-frozen for biochemical studies and storage. For measuring total content of proteins in muscle, the tissue was chopped into small pieces in RIPA lysis buffer (Santa Cruz Biotechnology, Dallas, TX, USA) containing protease and phosphatase inhibitors, and homogenized using a Polytron disrupter. The homogenate was centrifuged at 13000 g for 10 min and supernatant aliquots were stored in liquid nitrogen.

The microsomal fraction was prepared as described in *Perez et al., 2005*. Protein content was quantified by the BCA assay (Thermo-Fisher Scientific, Waltham, MA, USA). Proteins were separated by SDS–polyacrylamide gel electrophoresis, using 26-well pre cast gels (Criterion TGX, Bio-Rad, Hercules, CA, USA), which enable separation in a broad range of molecular weights, and transferred to a nitrocellulose membrane (Bio-Rad). Membranes were blocked at 4.5% blotting grade (Bio-Rad) in PBS and incubated with the primary antibody overnight at 4°C. Thereafter they were washed in PBS containing 0.1% Tween 20 and incubated in horseradish peroxidase–conjugated anti-mouse, anti-rabbit or anti-sheep secondary (Invitrogen, Carlsbad, CA, USA) for 1 hr at room temperature. The blot signals were developed with chemiluminescent substrate (Millipore, Burlington, MA, USA) and detected using the Syngene PXi system (Syngene USA Inc, Frederick, Md, USA). The figures represent 'positive' images (light intensity) in units of convenience on a 12 bit scale.

### Detecting multiple proteins in the same blot

For quantitative comparisons, multiple proteins were detected from the same nitrocellulose membrane after transference from gels. For this purpose, before applying antibodies the membrane was cut into strips centered on the protein of interest across the membrane's entire width, including the weight markers. The strips locations were defined with specific antibodies on full-length membranes. *Figure 2—figure supplement 3* illustrates the procedure.

Quantitative analysis of Western blots was done with a custom application (written in the IDL platform) that combined information in the blot and the source gel. This method had less variance than the commercial (Syngene) software tools. The content of interest was measured in the blot (*Figure 2A*) by the signal mass within a rectangle that enclosed the protein band (area a) above a background (measured in area b). A normalization factor, quantifying the sample deposited in the lane, was computed on the gel (*Figure 2B*) as the average signal in a large area of the corresponding lane (a) above background (b). Excluding from this area the region corresponding to the range of molecular weights of the proteins of interest improves slightly the sensitivity, especially in the analysis of microsomes. *Video 1* is a demonstration of this method.

## Immunostaining of human myofibers

Immunofluorescence examinations used thin myofiber bundles dissected from muscle biopsies. Bundles were mounted moderately stretched in relaxing solution, on Sylgard-coated dishes. Relaxing solution was replaced by fixative containing 4% PFA for 20 min. Bundles were transferred to a 24-well plate and washed three times for 10 min in PBS, then permeabilized with 0.1% Triton X-100 (Sigma) for 30 min at room temperature and blocked in 5% goat serum (Sigma) with slow agitation for 1 hr. The primary antibody was applied overnight at 4°C with agitation, followed by 3 PBS washes for 10 min. Fluorescent secondary antibody was applied for 2 hr at room temperature. Dehydrated bundles were mounted with Prolong Diamond anti-fade medium (Thermo-Fisher).

## High-Resolution imaging of fluorescence

### For immunostained human myofibers

Confocal imaging used a FluoView 1000 scanner (Olympus, Tokyo, Japan) or, for high resolution, a Falcon SP8 (Leica Microsystems, Wetzlar, Germany), both with 1.2 numerical aperture, water-immersion, 63x objectives. In the SP8, resolution was enhanced by high sensitivity hybrid GaAsP detectors (HyD, Leica), which allowed low intensity illumination for image averaging with minimum bleaching, optimal confocal pinhole size, collection of light in extended ranges and acquisition of *z*-stacks at oversampled *x–y–z* intervals. Most cells were dually stained and correspondingly monitored at (excitation/emission) (488/500–550 nm) and (555/570–620 nm). The stacks were acquired starting nearest the objective, at or closely outside the lower surface of the myofiber. The stacks consisted of 40 *x-y* images at 120 nm *z* separation and 60 nm *x-y* pixel size or, for high resolution imaging, 20 *x-y* images at 120 nm *z* and 36 nm *x-y* pixel size. Dual images were interleaved line by line.

### For fluorescently tagged mice myofibers

A DNA vector for the muscle isozyme of glycogen phosphorylase fused with GFP (GP-GFP) was transfected into paw muscles of adult mice (*Pouvreau et al., 2007*). Small fiber bundles of *Flexor Digitorum Brevis* (FDB) muscles expressing the protein were imaged stretched in a custom chamber (*Manno et al., 2016*). Imaging used a TCS SP2 confocal scanner (Leica). For colocalization studies, bundles were stained with the mitochondrial marker TMRE excited at 543 nm and imaged between 560 and 600 nm.

## Quantitative image analysis

The stacks of dual-stained images were first corrected for device 'bleed through' and spectral overlap. After background subtraction the fluorescence in dual measurements ($F_1$ and $F_2$) is

$$F_1 = A_1 + b_{21}A_2$$
$$F_2 = A_2 + b_{12}A_1 \tag{1}$$

Where $A_i$ is the contribution from marker $i$ (1 or 2), $b_{21}$ is the ratio between the fluorescence $F_1$ measured in the absence of marker 1, divided by the simultaneously measured $F_2$ and $b_{12}$ is the cross-coefficient determined in the opposite single-marker situation. From *Equation 1*:

$$A_1 = (F_1 - b_{21}F_2)/(1 - b_{21}b_{12})$$
$$A_2 = (F_2 - b_{12}F_1)/(1 - b_{21}b_{12}) \tag{2}$$

Corrected image stacks were then deblurred by a constrained iterative deconvolution algorithm that used all images in the stack (*Agard et al., 1989*; *Voort and Strasters, 1995*) with a point spread function determined in our microscopes. After deblurring, the separation effectively resolved in the *x-y* plane was approximately 0.25 μm for the FluoView and SP2 imagers, and approximately 0.1 μm for the SP8 (*Figure 5—figure supplement 2*). Representation or 'rendering' in 3-D (*Figure 5*) used the 'Simulated Fluorescence Process' (*Messerli et al., 1993*) applied to the full deblurred stack.

## Approaching the distribution of un-phosphorylated GP

Immunofluorescence images of phosphorylated glycogen phosphorylase, $F'_{GPa}$, and all-forms GP, $F_{GP}$, were used to derive a putative image of the *apo* form, $F_{GPb}$, as follows.

$F_{GP}$ is proportional to the quantity of protein. Assuming that the antibody is insensitive to phosphorylation of GP:

$$F_{GP} = B(GPa + GPb) \equiv F_{GPa} + F_{GPb} \tag{3}$$

*GPa* and *GPb* represent local densities of the respective proteins, *B* is the proportionality constant that links the fluorescence reported by the all-forms antibody to these densities. The second equality separates two additive contributions to the fluorescence. The fluorescence of *GPa*, reported by its specific antibody, satisfies a similar formula, with a different proportionality factor:

$$F'_{GPa} = CGPa \tag{4}$$

Therefore the fluorescence of the *apo* form is:

$$F_{GPb} = F_{GP} - F_{GPa} = F_{GP} - (B/C)F'_{GPa} \tag{5}$$

Therefore, the fluorescence of GP*b* is the difference between the all-GP fluorescence and that reported by the anti-GP*a* antibody multiplied by a constant. *Equation 5* will be applied to images of various dimensions.

## Colocalization analysis

Was carried out with custom cross-correlation algorithms implemented in the IDL programming environment (Harris Geospatiale, Paris, France), complemented with analysis by JACoP (*Bolte and Cordelières, 2006*), a plugin of ImageJ. Both are available at https://imagej.nih.gov/ij/.

## Glycogen analysis

The concentration of glycogen in muscle microsomal fractions was determined with colorimetric assay kit MAK016 (Sigma), using a microplate reader at 570 nm. The method provides for adding variable volumes of extract for a final content of 1 µg of protein per well. The readout, an absorbance, is then corrected for the variable quantity of sucrose added with the extraction buffer. Final content is evaluated as µg/µg of protein.

We tested whether the temperature at which the tissue was kept in the 18–24 hr interval between biopsy — in Toronto — and analysis — in Chicago — affected the results. Six samples from each of two patients, three of each kept frozen and the others kept at the standard 4°C, were analyzed. The frozen samples from one patient had on average 13% more glycogen in microsomes (p = 0.07). The difference in the second individual was 4%, in the same direction (p = 0.27).

## Statistics

Differences between two categories, the MHN and MHS, were tested for significance by the *t*-test when there was evidence of normality and equal variance in both groups. The non-parametric Mann-Whitney *u* test was used otherwise. Differences were considered significant at $p \leq 0.05$ in the two tails of the null effect distribution. To evaluate hypotheses of causation we quantified pairwise correlations, using the Pearson correlation coefficient *r*. The significance of the *r* value was calculated with the variable

$$t = r\sqrt{n}/\left(1 - r^2\right) \tag{6}$$

which in the null hypothesis has a Student's *t* distribution with $n - 2$ degrees of freedom (p. 466 of *Cramér, 1946*). An alternative variable appropriate for *r* > 0.5 (p. 467 of *Cramér, 1946*), is

$$v = \frac{\sqrt{n-3}}{2} log \frac{1+r}{1-r} \tag{7}$$

In the null hypothesis, *v* has a normal distribution with mean 0 and standard error 1.

### Multivariate statistics

Causation hypotheses were also tested through multivariate correlations, assessed by multivariate linear regression. The analysis was done in two ways, with equivalent results: (i) Direct: one independent variable ('*x*', typically the content of kinase PhK) as linearly linked to multiple dependent variables ('*y*': including GP, GP$_a$, GDE, GLUT4, other proteins and Glycogen). (ii) Inverse: one *y* (typically PhK) assumed to depend linearly on multiple *x* variables. The outputs of the analysis are the multivariate coefficient *r*, with statistical properties defined by *Equations 6 and 7*, a collective *F* value for *F*-tests of hypotheses, pairwise regression coefficients that quantify the dependence of one variable on the others in the linear hypothesis, and error estimates for these coefficients, which are used to test null hypotheses. Multiple regression was calculated using STATA (Statacorp, College Station, TX, USA), which also performs the *F* tests. Additional rationale, description and results of multivariate regression analysis are in Appendix 3.

## Sample size

It was defined in two stages: First we used samples from 14 or 15 patients (7 CHCT-positive and 7 or 8 negative) for establishing the methods. The sample size in this case was set by availability of biopsies and size of the available electrophoresis chamber (15 lane — 7 positives and 7 or eight negatives was the maximum size of two approximately equal size groups that could be compared in the same gel). Examples of work with 15-lane gels are in Appendices 1 and 3.

After the methods were optimized and aiming for higher sensitivity, we used 25-lane chambers —which recommended waiting for additional patients until the chambers could be used at full capacity. This led to the use of 12 positive + 13 negative muscles. It was not feasible to optimize sample size for a desired power. (Achieving greater sensitivity and power would have required unacceptable delays, as patients are recruited at the frequency of at most one per week). By an evaluation of probable error, it is argued in the next section that the sample size was generally sufficient to detect differences of 10% or greater in the measured protein contents.

## Replicates and probable error

Samples from different patients are bio-replicates of the MHS condition. Production of bio-replicates in an individual patient would entail multiple gels and blots of each protein, which was precluded in general by the limited material derived from a single biopsy. Therefore, most data reported here emerge from single gels and their derived blots.

To evaluate the probable error in the single gel determinations, we run a gel with three replicates (lanes) for each of 4 patients chosen because their large biopsies provided sufficient material. The gel is illustrated with Appendix 4. On a Western blot of GP$a$ from this gel we performed multiple evaluations (n = 5 'technical replicates') of normalized GP$a$ in all lanes. A simple analysis of variance then allowed calculation of expected errors in individual measures. The total variance —mean square deviation computed individually for each patient and averaged over patients — was 0.0303 when expressed as a fraction of the square of the means. The 'technical variance', that is the mean square deviation of the technical replications from the means in individual lanes measured repeatedly, was 0.0026. This is the component of the variance that can be reduced by repeated evaluations. The square root of the irreducible component, 0.166, is the SD that quantifies the probable error of the single measurement used in the study. Therefore, differences of less than 33% (2 × SD) between individuals cannot be assigned significance. When averaged over our sample of 12 or 13 (MHS or MHN) individuals, the probable error is reduced to ~9% of the measured value. The differences reported as significant in the study are much greater than 9%.

## Study approval

Following approval by the institutional Research Ethics Board of Toronto General Hospital (TGH), informed consents were obtained from all patients who underwent the CHCT. The consent, also approved by the Institutional Review Board of Rush University, included use of biopsies for functional studies, imaging and cell culture. Ethical aspects of the animal studies were approved by the IACUC of Rush University.

## Acknowledgements

Work supported by grants from the National Institutes of Health, USA. R01AR071381 (to ER, SR and M Fill, Rush University), R01AR072602 (to ER, S L Hamilton, S Jung and F Horrigan, Baylor College of Medicine), both from the National Institute of Arthritis and Musculoskeletal and Skin Diseases. R01GM111254 (to ER and C-H Kang, Washington State University) from the National Institute of General Medical Sciences. S1055024707 (to ER and others) from the National Center for Research Resources. We are grateful to Clara Franzini-Armstrong (Univ. Pennsylvania) for advice on interpretation of images and the gift of the EM image included in *Figure 5—figure supplement 1*. We are grateful to Dr Susan Hamilton, Baylor College of Medicine, for critical reading of the manuscript.

## Additional information

### Funding

| Funder | Grant reference number | Author |
|---|---|---|
| National Institute of Arthritis and Musculoskeletal and Skin Diseases | R01AR071381 | Sheila Riazi Eduardo Rios |
| National Institute of Arthritis and Musculoskeletal and Skin Diseases | R01AR072602 | Eduardo Rios |
| National Institute of General Medical Sciences | R01GM111254 | Eduardo Rios |
| National Center for Research Resources | S1055024707 | Eduardo Rios |

The funders had no role in study design, data collection and interpretation, or the decision to submit the work for publication.

### Author contributions

Eshwar R Tammineni, Conceptualization, Resources, Data curation, Software, Formal analysis, Supervision, Funding acquisition, Validation, Investigation, Visualization, Methodology, Writing - original draft, Project administration, Writing - review and editing; Natalia Kraeva, Conceptualization, Resources, Formal analysis, Validation, Investigation, Methodology, Writing - original draft, Writing - review and editing; Lourdes Figueroa, Resources, Data curation, Investigation, Methodology; Carlo Manno, Resources, Data curation, Formal analysis, Investigation, Visualization, Methodology; Carlos A Ibarra, Data curation, Formal analysis, Investigation, Writing - review and editing; Amira Klip, Conceptualization, Resources, Formal analysis, Investigation, Visualization, Methodology, Writing - original draft, Writing - review and editing; Sheila Riazi, Conceptualization, Resources, Data curation, Formal analysis, Supervision, Funding acquisition, Validation, Methodology, Writing - original draft, Project administration, Writing - review and editing; Eduardo Rios, Conceptualization, Resources, Data curation, Software, Formal analysis, Supervision, Funding acquisition, Validation, Visualization, Methodology, Writing - original draft, Project administration, Writing - review and editing

### Author ORCIDs

Carlos A Ibarra http://orcid.org/0000-0001-8898-6772
Amira Klip http://orcid.org/0000-0001-7906-0302
Eduardo Rios https://orcid.org/0000-0003-0985-8997

### Ethics

Human subjects: Following approval by the institutional Research Ethics Board of Toronto General Hospital (TGH), informed consents were obtained from all patients who underwent the CHCT. The consent, also approved by the Institutional Review Board of Rush University, included use of biopsies for functional studies, imaging and cell culture.

Animal experimentation: This study was performed in strict accordance with the recommendations in the Guide for the Care and Use of Laboratory Animals of the National Institutes of Health. All of the animals were handled according to approved institutional animal care and use committee (IACUC) protocols of Rush University (# 17-035, 16-091 and 18-065). All surgery was carried out on animals previously euthanized by methods approved under said protocols. Every effort was made to minimize stress and suffering.

### Decision letter and Author response

Decision letter https://doi.org/10.7554/eLife.53999.sa1
Author response https://doi.org/10.7554/eLife.53999.sa2

## Additional files

### Supplementary files
- Source data 1. Raw data in multi-page worksheet.
- Supplementary file 1. Mass spectrometry report table.
- Transparent reporting form

### Data availability

All data generated or analysed during this study are included in the manuscript and supporting files. Source data files have been provided for all figures and tables in a multi-sheet Excel file.

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

## Appendix 1

### Proteomics Report

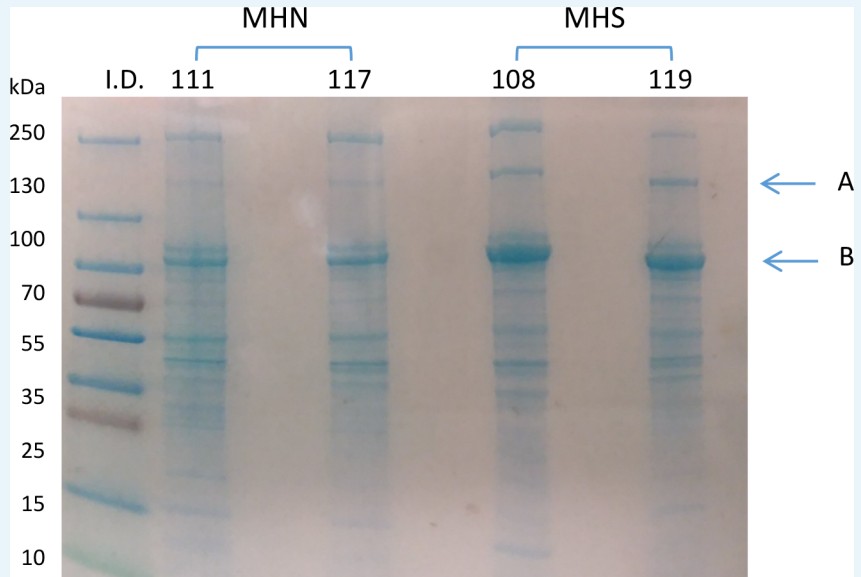

**Appendix 1—figure 1.** 20 µg of microsomal extracts from muscle specimens of 2 MHN and 2 MHS patients identified in the figure were run by SDS PAGE. The gel was stained with Gel Code Blue (Thermo Scientific). Bands of 170 and 100 KDa (A and B in figure) were excised for liquid chromatography-mass spectrometry (LC–MS) and further processed for protein identification at Rush University Proteomics Facility. The report includes an account of parameters of the dominant protein (*Appendix 1—table 1*) and a table of all identified peptides, presented as *Supplementary file 1*. The analysis was consistent with the presence of multiple proteins in both gel bands. The peptide count and protein sequence coverage identified the dominant proteins as glycogen debranching enzyme (GDE), *Homo sapiens*, molecular weight 175 KDa and glycogen phosphorylase muscle form (GP), *Homo sapiens*, with 97 KDa molecular weight. Additionally, SERCA1 was found in abundance in band B.

**Appendix 1—table 1.** Col. 1, dominant protein in bands A (GDE) or B (GP). Col. 2, individual patient identifier. Cols. 3 and 4, number of unique peptides and their sequence coverage, corresponding to the protein named in the first column. Numbers and sequence coverage data are consistent with predominant presence of GDE in band A and GP and SERCA1 in band B. The numbers of GDE and GP peptides in the respective bands are greater in MHS patients, which suggests a greater content that was confirmed with quantitative immunoblotting.

| GDE | Patient ID | unique peptide count | protein sequence coverage (%) |
|---|---|---|---|
| | 111(MHN) | 10 | 6.4 |
| | 117(MHN) | 5 | 3.5 |
| | 108(MHS) | 18 | 13 |
| | 119(MHS) | 23 | 16 |
| GP | Patient ID | unique peptide count | protein sequence coverage (%) |
| | 111(MHN) | 46 | 57 |
| | 117(MHN) | 49 | 59 |
| | 108(MHS) | 68 | 69 |
| | 119(MHS) | 62 | 65 |

## Appendix 2

### Biochemical analysis for an alternative group of patients

Muscle tissue from a first group of 26 patients, 12 MHS and 14 MHN, was subjected to Western blot analyses of six proteins and quantification of glycogen using standard techniques, which were replaced by custom ones for deriving the data presented in the main article. The techniques included electrophoresis in a smaller 15-lane chamber (stained gel in figure), conventional normalization by stained GAPDH, and quantification by commercial software. The data in the main article instead used a 25-lane chamber (which permitted simultaneous study of 12 MHS and 13 MHN patients), normalization by the signal from a large set of proteins and quantification by a custom software (demonstrated in *Video 1*). These first results are summarized in the table (*Appendix 2—table 1*). They show similar differences between MHS and MHN, but with lower statistical significance, due to greater dispersion and smaller numbers of patients compared.

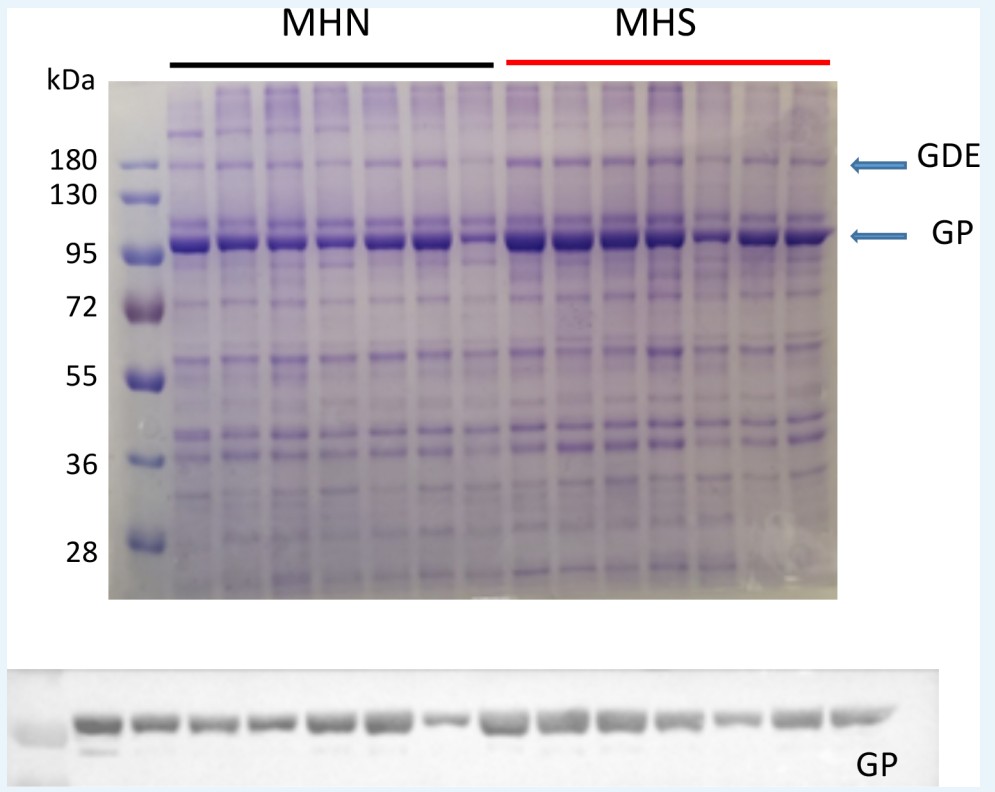

**Appendix 2—figure 1.** Example of the smaller gels used with this first group. In the example and in every gel prepared from microsomes, the excess protein at ~100 and~180 kDa is noticeable.

**Appendix 2—table 1.** Summary of biochemical analyses of alternative groups. Quantitative comparisons of 7 MHN (selected among 14) and 7 MHS (selected among 11) samples run on the same 15-lane electrophoresis gel. §Significant at $p \leq 0.05$ by two-tailed Student's $t$ test. *Significant at $p \leq 0.05$ by Mann-Whitney $u$ test.

| Species | Fraction | MHN | | | | MHS | | | | P |
|---------|----------|------|--------|--------|----|------|--------|--------|----|---|
| | | Mean | Median | S.E.M. | N | Mean | Median | S.E.M. | N | |
| GP | mc | 0.65 | 0.63 | 0.052 | 14 | 1.39 | 1.28 | 0.175 | 11 | $<10^{-3*}$ |
| | wm | 0.90 | 0.89 | 0.036 | 7 | 0.94 | 1.28 | 0.061 | 7 | 0.450 |
| GPa | wm | 1.30 | 1.06 | 0.176 | 7 | 1.69 | 1.64 | 0.121 | 7 | 0.027* |
| GPa/GP | wm | 1.43 | 1.20 | 0.170 | 7 | 1.92 | 1.83 | 0.102 | 7 | 0.069 |
| PhK | mc | 0.49 | 0.39 | 0.107 | 7 | 1.07 | 0.95 | 0.230 | 7 | $<10^{-3*}$ |
| | wm | 1.04 | 1.00 | 0.214 | 7 | 1.89 | 1.87 | 0.267 | 7 | 0.029§ |
| GDE | mc | 0.60 | 0.55 | 0.087 | 7 | 1.06 | 1.05 | 0.091 | 7 | 0.004* |
| GS | mc | 7.63 | 8.94 | 0.857 | 7 | 8.09 | 8.02 | 0.574 | 7 | 0.66 |
| GSa/GS | mc | 0.82 | 0.82 | 0.16 | 7 | 1.17 | 1.17 | 0.075 | 7 | 0.13 |
| Glycogen | mc | 1.93 | 1.97 | 0.074 | 7 | 0.93 | 0.65 | 0.307 | 7 | 0.008§ |

**Appendix 3**

## Multivariate statistical analysis

It considers jointly the correlation of multiple measured variables with a putative causation variable. The joint consideration allows 'evidence synthesis', which potentially gives significance to correlations that in a 2-variable analysis may not reach p < 0.05. (**Andrinopoulou et al., 2014**; **Rizopoulos and Lesaffre, 2014**). This is an example of multiple linear regression to test the likelihood that the increased presence of PhK causes the increase in GP in the microsomal fraction, and GPa in the whole muscle extract. The data are given in the table (**Appendix 3—table 1**) in the units used in the main text, with the exception of glycogen, which is included in mg/l of extract.

**Appendix 3—table 1.** Data for multivariate analysis. 'whm '– whole tissue extract; 'mic' – microsomal fraction. The top set of data are derived from MHN patients and the bottom set from the MHS, but the analysis is blind to this distinction. Analysis by Multiple Regression, which inverts the problem, finding the best fit of PhK (3$^{rd}$ column) as a linear function of the quantities in columns 1 and 2. The following are the outputs of the analysis:

| GPa whm | GPa mic | PhK mic | GDE mic | GLUT4 mic | Glycogen mic |
|---|---|---|---|---|---|
| 2.07 | 2.75 | 0.54 | 2.38 | 0.75 | 14.78 |
| 1.42 | 3.39 | 0.29 | 1.62 | 1.30 | 16.24 |
| 1.79 | 4.00 | 0.15 | 2.04 | 0.24 | 12.81 |
| 1.10 | 2.62 | 0.55 | 0.77 | 1.23 | 23.65 |
| 1.05 | 3.10 | 0.76 | 2.77 | 2.23 | 26.27 |
| 1.05 | 1.85 | 0.23 | 0.70 | 1.84 | 8.50 |
| 1.18 | 2.00 | 0.60 | 0.80 | 1.94 | 21.67 |
| 1.35 | 3.05 | 0.35 | 0.47 | 0.77 | 24.71 |
| 0.90 | 2.78 | 0.59 | 0.38 | 1.62 | 7.22 |
| 1.41 | 3.58 | 0.46 | 2.33 | 1.32 | 16.26 |
| 1.27 | 6.16 | 0.22 | 0.37 | 0.56 | 12.73 |
| 1.11 | 8.25 | 0.31 | 1.85 | 0.13 | 19.03 |
| 1.53 | 6.07 | 0.87 | 0.95 | 0.43 | 6.31 |
|  |  |  |  |  |  |
| 1.62 | 5.07 | 1.33 | 1.03 | 1.45 | 11.72 |
| 1.16 | 3.48 | 0.67 | 0.81 | 0.46 | 14.52 |
| 1.55 | 2.83 | 0.08 | 1.81 | 0.68 | 9.48 |
| 2.04 | 4.04 | 0.93 | 0.91 | 0.79 | 9.08 |
| 2.13 | 4.46 | 1.87 | 0.88 | 0.58 | 4.53 |
| 1.61 | 4.35 | 1.16 | 1.03 | 0.54 | 11.06 |
| 2.03 | 4.19 | 0.48 | 1.14 | 0.17 | 8.81 |
| 3.36 | 8.32 | 1.06 | 2.32 | 0.42 | 10.68 |
| 2.69 | 5.51 | 1.12 | 0.93 | 0.92 | 14.64 |
| 2.59 | 15.20 | 3.08 | 4.10 | 0.44 | 5.59 |
| 3.16 | 15.00 | 2.37 | 1.59 | 0.25 | 8.81 |
| 2.71 | 5.52 | 1.92 | 2.73 | 1.07 | 4.86 |

Multiple correlation coefficient $r = 0.788$. $r^2 = 0.62$. Regression sum of squares (or Explained SS) $ESS = 8.13$. Residual sum of squares, $RSS = 4.97$. $n$ (cases) = 25; $k$

(variables) = 3. From these, an *F*-test of the null hypothesis was done by computing $F (=ESS(n\text{-}k)/RSS(k\text{-}1))$ = 17.2 and p = 3.83 $10^{-5}$.

Using the same procedure, an *F* test of the 3-variate correlation between PhK, GP*a* in microsomes and GDE gave a p = 5 $10^{-4}$, between PhK, GP*a* and Glycogen a p = 5.2 $10^{-5}$ and between PhK, GP*a* and GLUT4, p = 4.9 $10^{-5}$. This approach can be extended to more dimensions. Thus a 4-variate correlation of PhK, GDE, GLUT4 and Glycogen yields p = 0.010, improving greatly on the significance of the paired correlations between PhK and the other three variables (*p* values of paired correlations are in main article *Table 2*).

Multiple regression was calculated by the Excel Add-In 'Data Analysis' and by 'STATA' (Statacorp, College Station, TX, USA), with the same results. STATA also performs the *F* test.

## Appendix 4

### An estimation of probable error in quantification of Western blots

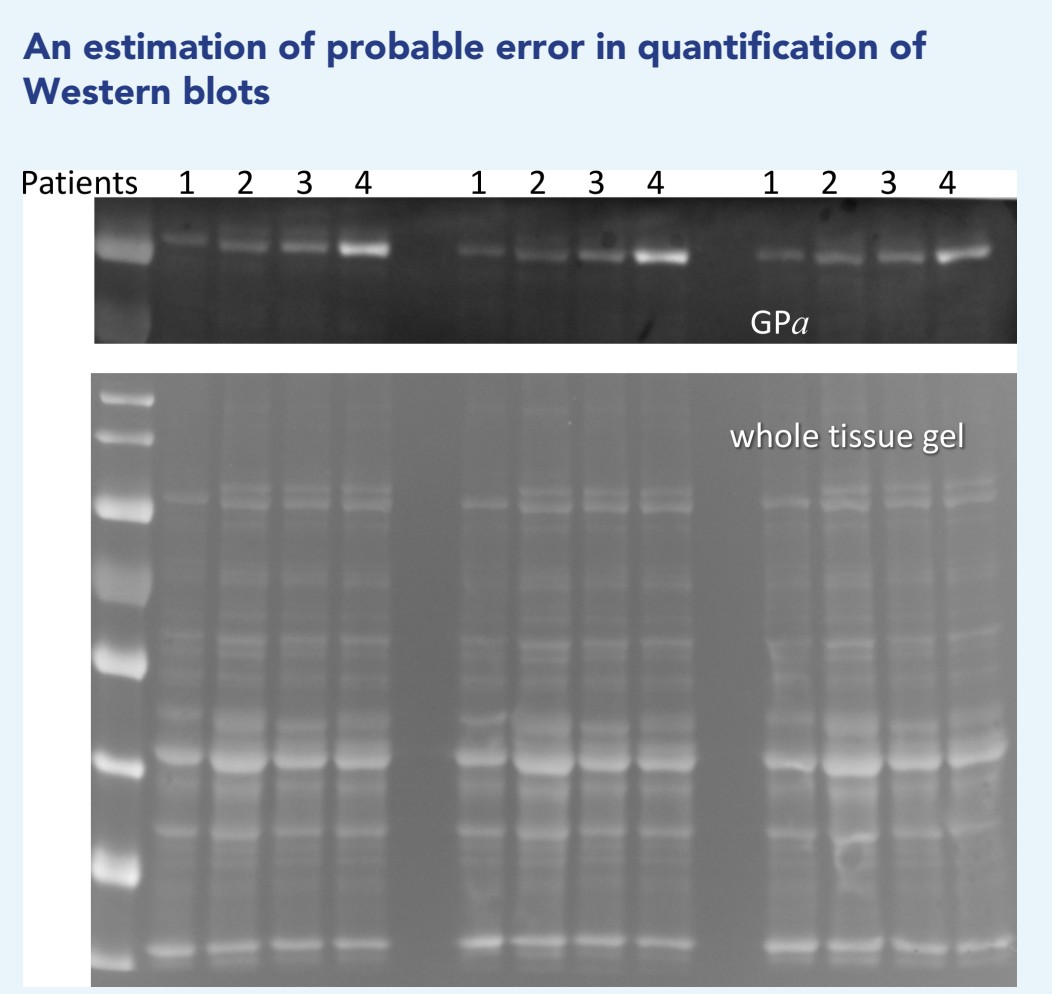

**Appendix 4—figure 1.** Western blot of GPa for whole cell extracts of 4 patients. Material from the same individual cell extract were loaded in three lanes as indicated. 5 repeated measurements were carried out for every lane. See Methods/Statistics/Replicates and Probable Error for additional information.

