## [Decision Letter]

**Acceptance summary:**

This study provides a significant advance regarding how muscle glycogen metabolism is altered in Malignant Hyperthermia Susceptibility patients. The proposal that a specific Ca-dependent pathway involved in glycogen metabolism is altered in Malignant Hyperthermia Susceptibility patients is novel.

**Decision letter after peer review:**

Thank you for submitting your article "A wide-ranging alteration of glucose metabolism in muscle of patients with defective excitation-contraction coupling" for consideration by *eLife*. Your article has been reviewed by three peer reviewers, and the evaluation has been overseen by a Reviewing Editor and Richard Aldrich as the Senior Editor. The following individuals involved in review of your submission have agreed to reveal their identity: Enrique Jaimovich (Reviewer #3).

The reviewers have discussed the reviews with one another and the Reviewing Editor has drafted this decision to help you prepare a revised submission.

Summary:

This study investigates the mechanism for the higher incidence of hyperglycemia in patients susceptible to malignant hyperthermia (MHS) reported previously (Altamirano et al., 2019). By analyzing muscle samples from MHN and MHS patients, the authors propose a pathway driven by elevated cytosolic resting [Ca^2+^] in MHS, which activates Ca^2+^-dependent phosphorylase kinase (PhK) to phosphorylate glycogen phosphorylase (GP) and glycogen synthase (GS) to shift the glucose-glycogen balance towards glycogenolysis. Elegant immuolocalization studies indicate that phopsphorylation of GS (GSα) results in translocation of the enzyme from the Z-line to the terminal SR. The increase in gycogenolysis, coupled with a reduction in glucose 4 transporter (Glut4) expression, is proposed to lead to the hyperglycemia observed in these patients.

Essential revisions:

1) Most of the data in this study were obtained using a "recent cohort" of 12 MHS and 13 MHN patients. While analysis of a larger cohort of 560 patients (329 MHS and 231 MHN) termed the "legacy cohort" provide clear evidence of increased hyperglycemia in MHS patients, average fasting glycaemia (FBS) MHS patients of the recent cohort was not significantly elevated compared with the MHN patients (Figure 1—figure supplement 1). This complicates interpretation and inferences obtained regarding the relevance for hyperglycemia of changes observed in the more limited recent cohort. The authors did resolve an increase in FBS (Table 1) between a subgroup of the MHS patients that have the most pronounced change in GPα and GPα/GP ratio and a subgroup of the MHN patients with the lowest levels of these enzymes. However, the relevance of this selected subgroup analysis is unclear, especially since a similar sub-cohort analysis was not conducted between MHS patients with low GPα and MHN patients with high GPα. A correlation analysis between FBS and levels of GPα in microsomes using all the patients in the cohort might be informative.

2) A recent study by this group (Figueroa et al., 2019) reported that MHS in the legacy cohort can be divided into patients that responded to both halothane and caffeine in CHCT test (HS) or those that responded to only halothane (HH). Importantly, this study reported that myotubes derived from patients in the HH group exhibited significantly higher cytosolic resting [Ca^2+^] levels compared with both myotubes from patients in the MHN and HS groups. In fact, resting [Ca^2+^] levels were not different significantly between MHN and HS. Were the 12 MHS individuals used in the recent cohort from the HS or HH subgroups? Since the authors propose that elevated cytosolic [Ca^2+^] drives changes in GP phosphorylation, subcellular localization and hyperglycemia, it would be informative to compare FBS (and levels of GPα in microsomes) between MHN, HH, and HS patients.

3) The elevation of resting [Ca^2+^] used in the Ca^2+^ challenge experiments described in Figure 7 (300-500nM) are outside of the physiological range. For example, Figueroa et al., 2019 reported only a modest increase in resting cytosolic [Ca^2+^] in myotubes derived from HH patients (from ~100 nM in MHN to ~130nM in HH). As an alternative to consider, Cully et al., 2019 (PMID: 30038012), reported that increased Ca^2+^ leak in muscle fibers from MHS patients results in a significantly larger increase in [Ca^2+^] within the junctional space compared to global cytosolic [Ca^2+^] levels. Thus, local Ca^2+^ levels within the junctional space may be large enough to drive PhK-dependent phosphorylation of GP. In this case, it would be important to determine if PhK localizes to the A-I band junction where it would be exposed to the higher junctional Ca^2+^ levels created by increased RYR1 Ca^2+^ leak. In this case, elevated nanodomain junctional Ca^2+^ resulting from RYR1 Ca^2+^ leak could act to phosphorylate GP and trap it close to the SR. This possibility should be considered in the Discussion section.

4) The authors conclude that MHS patients exhibit hyperglycemia as a result of enhanced glycogen breakdown. However, the level of glycogen reported is extremely low even in MHN, compared to previously studies. It is acknowledged that is from microsomes, but still ~20mg/liter is extremely low. The authors obtain muscle biopsies from gracilis, hence glycogen in whole muscle homogenate would give more of an indication if the obtained glycogen levels appear physiological relevant. Please clarify.

5) The glycogen experiments are, if understood correctly, not performed on flash frozen (or freeze-dried) muscle but instead after the muscles have been floating around in DMEM media overnight, which potentially could affect the glycogen content. Please clarify.

6) There are no comparisons to healthy individuals, to establish some of the GP, GPα /GLUT4 etc. levels as well as glycogen content in healthy subjects would emphasize if the findings are higher/lower/similar as compared to a healthy cohort. This needs to be addressed and clarified by the authors to understand the impact of the findings.

---

## [Author Response]

Essential revisions:1) Most of the data in this study were obtained using a "recent cohort" of 12 MHS and 13 MHN patients. While analysis of a larger cohort of 560 patients (329 MHS and 231 MHN) termed the "legacy cohort" provide clear evidence of increased hyperglycemia in MHS patients, average fasting glycaemia (FBS) MHS patients of the recent cohort was not significantly elevated compared with the MHN patients (Figure 1—figure supplement 1). This complicates interpretation and inferences obtained regarding the relevance for hyperglycemia of changes observed in the more limited recent cohort. The authors did resolve an increase in FBS (Table 1) between a subgroup of the MHS patients that have the most pronounced change in GPα and GPα/GP ratio and a subgroup of the MHN patients with the lowest levels of these enzymes. However, the relevance of this selected subgroup analysis is unclear, especially since a similar sub-cohort analysis was not conducted between MHS patients with low GPα and MHN patients with high GPα. A correlation analysis between FBS and levels of GPα in microsomes using all the patients in the cohort might be informative.

We calculated the correlation between FBS and three glycogen proteins with the following results: FBS vs. phosphorylation ratio of GP (GP*a*/GP) in whole muscle, *r* = 0.58, *p* of no correlation, 0.002. FBS vs. GP*a* in whole muscle, *r* = 0.49; *p* = 0.01; FBS vs. GP*a* in microsomes, *r* = 0.29; *p* = 0.15). The plot of FBS vs. phosphorylation ratio of GP in whole muscle is in new Figure 10. The results are consistent with the correlations already demonstrated in the original. The superior correlation with the phosphorylation in whole tissue is in line with the finding that the most robust results are obtained with the whole muscle.

2) A recent study by this group (Figueroa et al., 2019) reported that MHS in the legacy cohort can be divided into patients that responded to both halothane and caffeine in CHCT test (HS) or those that responded to only halothane (HH). Importantly, this study reported that myotubes derived from patients in the HH group exhibited significantly higher cytosolic resting [Ca^2+^] levels compared with both myotubes from patients in the MHN and HS groups. In fact, resting [Ca^2+^] levels were not different significantly between MHN and HS. Were the 12 MHS individuals used in the recent cohort from the HS or HH subgroups? Since the authors propose that elevated cytosolic [Ca^2+^] drives changes in GP phosphorylation, subcellular localization and hyperglycemia, it would be informative to compare FBS (and levels of GPα in microsomes) between MHN, HH, and HS patients.

This is a desirable analysis. However, it is not informative with the data at hand, as the 12 “MHS” patients fully studied include one “HS” and 11 “HH” patients. The unequal distribution was determined solely by availability of samples. Anecdotally, this HS is one of the “Metabolically compromised” patients already mentioned in the original manuscript. (Its coordinates in the figure are: GP*a*/GP = 0.65, FBS = 6.1 mM).

3) The elevation of resting [Ca^2+^] used in the Ca^2+^ challenge experiments described in Figure 7 (300-500nM) are outside of the physiological range. For example, Figueroa et al., 2019 reported only a modest increase in resting cytosolic [Ca^2+^] in myotubes derived from HH patients (from ~100 nM in MHN to ~130nM in HH). As an alternative to consider, Cully et al., 2019 (PMID: 30038012), reported that increased Ca^2+^ leak in muscle fibers from MHS patients results in a significantly larger increase in [Ca^2+^] within the junctional space compared to global cytosolic [Ca^2+^] levels. Thus, local Ca^2+^ levels within the junctional space may be large enough to drive PhK-dependent phosphorylation of GP. In this case, it would be important to determine if PhK localizes to the A-I band junction where it would be exposed to the higher junctional Ca^2+^ levels created by increased RYR1 Ca^2+^ leak. In this case, elevated nanodomain junctional Ca^2+^ resulting from RYR1 Ca^2+^ leak could act to phosphorylate GP and trap it close to the SR. This possibility should be considered in the Discussion section.

The alternative proposed by the reviewer, that PhK is activated at junctions by local high Ca, is worth of further study and is now mentioned, together with the results of Cully et al. However, we have two strong reasons for not including further examination of PhK phosphorylation in the present study.

1) We have already noted that the present observations pose major conundrums. Specifically about PhK we wrote: “…PhK increases in the microsomes but remains constant in the whole tissue extract. The change thus appears to be a migration of the kinase towards the membrane fraction. The functional implications of PhK migration are intriguing, considering that GP*b* – its substrate – remains at the I band, near the Z disk (Figure 4 and (1,2)), while GP*a* appears near the SR. The spatial aspects of the interactions between the kinase and its substrates remain to be defined”. In short, PhK (re)distribution and interactions are intriguing topics, which won’t be clarified without major work.

2) The way ahead would seem to be via an imaging study of PhK’s apo and phospho forms, similar to that done for GP. However, this is not possible because there is to our knowledge no suitable pair of antibodies. For the above reasons, we believe that pursuing this line would extend the scope and enlarge the size and difficulties of the study beyond what we consider feasible at this time.

4) The authors conclude that MHS patients exhibit hyperglycemia as a result of enhanced glycogen breakdown. However, the level of glycogen reported is extremely low even in MHN, compared to previously studies. It is acknowledged that is from microsomes, but still ~20mg/liter is extremely low. The authors obtain muscle biopsies from gracilis, hence glycogen in whole muscle homogenate would give more of an indication if the obtained glycogen levels appear physiological relevant. Please clarify.

We chose to quantify glycogen in microsomes based on the evidence that a (large) majority of the glycogen identified in EM images is in the intermyofibrillar spaces, and should migrate with the membranes (microsomal) fraction. We refer readers to Figure 1C and D of Ørtenblad and Nielson (2015), a recent article that reviews quantitative aspects of the distribution of glycogen in human muscle. While a significant difference in glycogen content should be detectable in whole muscle as well, we thought that the potential error would be lower in the more concentrated preparation. Regardless of the scale of the result (more about this below), the difference between identically treated groups is highly significant.

To decide whether the quantity of glycogen found in the microsomal fraction is low, we realized that expressing the results per liter of preparation is adequate for the comparison but not in the absolute sense (it depends on dilutions intrinsic to the “kit” used). We have recast the results as µg glycogen/µg microsomal protein (g/g). The graphs and statistical calculations have been changed accordingly. In these units the means are 1.16 (MHN) and 0.72 (MHS) and the *p* of no difference (in the non-parametric test) continues to be 0.01. For comparison, Lees et al. (3) report 0.4 g per g of microsomal protein in rested rat muscles, a maximum that is drastically reduced by activity. Entman et al. (4) gives a comparable range of values in dog muscle, while Cuenda’s numbers are much lower (5).

In sum, our measured glycogen content is somewhat greater than comparable values we could find in the literature. Our choice to quantify glycogen in the microsomal fraction seems reasonable, given its preferential intermyofibrillar location. We have now expanded the justification of the study in Results.

5) The glycogen experiments are, if understood correctly, not performed on flash frozen (or freeze-dried) muscle but instead after the muscles have been floating around in DMEM media overnight, which potentially could affect the glycogen content. Please clarify.

The concern does not seem critical, as we report values from samples processed identically. To calibrate the difference associated with the time elapsed in solution, three dual samples from each of two individuals (#177 and 178, biopsied in February and March of this year) were analyzed. One part of each sample, “snapfrozen”, was kept at -20 °C and shipped in liquid N and the other “standard” was kept at ~4 °C since the time of excision. 4 grams of muscle from each patient were processed in total. For #711 the averages (SEM) are 1.11 (0.02) and 1.25 (0.05) respectively for snap-frozen and standard conditions. The difference, of ~12%, is not significant (*p* = 0.073). For #178 they were 1.84 (0.05) and 1.91 (0.03) respectively. The difference, of ~4%, is again not significant (*p* = 0.27). Averages of both sets are also not significantly different; however, the glycogen content was lower in most individual pieces of tissue kept frozen, which might reflect greater dephosphorylation of GP*a* in the samples shipped at 4°C.

We now include this description in Materials and methods section. A reference to this test is also in Results.

6) There are no comparisons to healthy individuals, to establish some of the GP, GPα/GLUT4 etc. levels as well as glycogen content in healthy subjects would emphasize if the findings are higher/lower/similar as compared to a healthy cohort. This needs to be addressed and clarified by the authors to understand the impact of the findings.

It is difficult to obtain large muscle biopsies from a sizable sample of healthy individuals. However, by all conventional criteria our MHN cohort appears as a suitable substitute for healthy individuals. None of our MHNs has signs of myopathies, or other muscle disease. With few exceptions, these are individuals tested only because of family history, who additionally have proven negative to relevant gene mutations.

References

1) Maruyama K, Kuroda M, Nonomura Y. Association of chicken pectoralis muscle phosphorylase with the Z-line and the M-line of myofibrils: comparison with “amorphin”, the amorphous component of the Z-line. Biochim Biophys Acta. 1985 Jun 10;829(2):229–37.

2) Chowrashi P, Mittal B, Sanger JM, Sanger JW. Amorphin is phosphorylase; phosphorylase is an alpha-actinin-binding protein. Cell Motil Cytoskeleton. 2002 Oct;53(2):125–35.

3) Lees SJ, Franks PD, Spangenburg EE, Williams JH. Glycogen and glycogen phosphorylase associated with sarcoplasmic reticulum: effects of fatiguing activity. J Appl Physiol Bethesda Md 1985. 2001 Oct;91(4):1638–44.

4) Entman ML, Keslensky SS, Chu A, Van Winkle WB. The sarcoplasmic reticulum-glycogenolytic complex in mammalian fast twitch skeletal muscle. Proposed in vitro counterpart of the contraction-activated glycogenolytic pool. J Biol Chem. 1980 Jul 10;255(13):6245–52.

5) Cuenda A, Henao F, Nogues M, Gutiérrez-Merino C. Quantification and removal of glycogen phosphorylase and other enzymes associated with sarcoplasmic reticulum membrane preparations. Biochim Biophys Acta. 1994 Aug 24;1194(1):35–43.